# Internal states as a source of subject-dependent movement variability are represented by large-scale brain networks

Macauley Smith Breault [1,2] ✉, Pierre Sacré [3], Zachary B. Fitzgerald[4], John T. Gale[5], Kathleen E. Cullen [2], Jorge A. González-Martínez[6] & Sridevi V. Sarma[2]

Humans' ability to adapt and learn relies on reflecting on past performance. These experiences form latent representations called internal states that induce movement variability that improves how we interact with our environment. Our study uncovered temporal dynamics and neural substrates of two states from ten subjects implanted with intracranial depth electrodes while they performed a goal-directed motor task with physical perturbations. We identified two internal states using state-space models: one tracking past errors and the other past perturbations. These states influenced reaction times and speed errors, revealing how subjects strategize from trial history. Using local field potentials from over 100 brain regions, we found large-scale brain networks such as the dorsal attention and default mode network modulate visuospatial attention based on recent performance and environmental feedback. Notably, these networks were more prominent in higher-performing subjects, emphasizing their role in improving motor performance by regulating movement variability through internal states.

Professional athletes represent the pinnacle of motor control and precision, but they too fall victim to slight variations in their movement. Movement variability is traditionally viewed as a byproduct of noise accumulated by the motor system[1]. However, there is emerging evidence that this variability is purposefully orchestrated by the brain to facilitate learning and adaptation[2–5]. For example, more movement variability through exploration leads to faster learning and better performance[6,7]. The decision to explore—as opposed to exploit—an environment to gather information to inform future behavior through learning lends itself naturally to movement variability. Not only does this information depend on the present, but also on internalized factors that account for the accumulation of past experience. These factors are commonly represented as a methodological construct of memory called internal states. For example, movement variability is

influenced by motivation[8,9], confidence[10,11], and emotion[12–14]. Future behavior is therefore the culmination of current information and internal states.

With their impact on behavior so apparent, it is surprising how ambiguous internal states are in motor control compared to other fields such as decision-making. To date, research into decision-making has used statistical models to explore relationships between behavioral variability and internal states[15–18] with the aim of finding evidence of the brain encoding states related to uncertainty[19], bias[20], trial history[21], and impulsivity[22]. Like decision-making, the goal of motor control is to produce actions that optimize outcomes in the presence of uncertainty[23]. Whether those actions are decisions or movements, variability and internal states are inherent to both. Therefore, we speculated that internal states are encoded in regions that are not

[1]Picower Institute for Learning and Memory, Massachusetts Institute of Technology, Cambridge, MA, USA. [2]Department of Biomedical Engineering, Johns Hopkins University, Baltimore, MD, USA. [3]Department of Electrical Engineering and Computer Science, School of Engineering, University of Liège, Liège, Belgium. [4]Department of Neurology, Feinberg School of Medicine, Northwestern University, Chicago, IL, USA. [5]DIXI Neurolab, Inc., Oxford, MI, USA. [6]Department of Neurological Surgery, University of Pittsburgh, Pittsburgh, PA, USA. ✉e-mail: breault@mit.edu

specific to motor control (i.e., nonmotor regions). Indeed, decision-making tasks that require movements find that regions involved with sensorimotor integration encode their internal states as opposed to motor regions[20,21]. In this context, an emerging consensus is that movement variability originates from both the planning[24] and execution phases[25] of movement. However, the gap in our understanding of motor control becomes apparent when one asks how internal states evolve, where they are encoded in the brain, and how they affect performance.

Two challenges need to be overcome to address these questions. The first challenge is determining the internal states based on behavior. To date, direct measures of the brain's internal states have remained elusive[26]. Internal states are not measurable biological phenomena, such as temperature. Researchers have tried to capture measures of such states using methods including self-reporting[27], galvanic skin conductance[28], heart rate variability[29,30], and pupil size[31]. However, these measures are context-dependent[32], vary between individuals[30], and can function with timescales on the order of minutes to compute[28,30], whereas internal states can change within seconds[33]. By comparison, decision-making studies rely on observable behavior such as reaction times, decisions, and outcomes to derive their internal states. Therefore, motor control studies would also be ideal for deriving internal states due to their abundant movement-related data. Even so, since these methods are all byproducts of the brain, the question arises as to why not measure internal states directly from the brain.

This leads to our second challenge, which is identifying where internal states are encoded in the brain. As previously mentioned, research on decision-making supports the view that internal states are encoded by diverse brain regions involving multiple systems (e.g., sensory and memory)[19–22]. Whole-brain imaging with high temporal resolution would be ideal for capturing diverse brain structures and rapidly evolving internal states. Most work in humans has used non-invasive neural imaging, such as functional Magnetic Resonance Imaging (fMRI). These studies report the occurrence of co-varying behavioral and neurological variability[2,4,34]. However, the limitations of the temporal resolution of fMRI[35], compounded by the limited space that subjects must perform a natural movement, make it difficult to link neural correlates of internal states to behavior[34]. What is needed is millisecond resolution with whole-brain coverage.

To address these challenges, we combined high-quality measurements of natural reaching movements with high-spatial and temporal resolution neural StereoElectroEncephaloGraphy (SEEG) recordings. Specifically, ten human subjects implanted with intracranial depth electrodes performed a simple motor task that elicited movement variability during planning and execution. We estimated two internal states using state-space models trained on measurable behavioral data: the error state accumulates based on past errors to convey overall performance and the perturbed state accumulates

based on past perturbations to convey environmental uncertainty. Adding these states improved our ability to estimate trial-by-trial reaction times and speed errors over using stimuli alone. Our approach also granted us access to latent terms that predicted subject performance and provided insight into subject strategy. Remarkably, we found neural evidence of the brain encoding each of the internal states in relatively distinct large-scale brain networks. Specifically, large-scale brain networks, such as the Dorsal Attention Network (DAN) and Default Network (DN), were linked to the error and perturbed state. We also have preliminary evidence linking the encoding strength and functional connectivity of these networks back to subject performance and strategy.

## Results

To investigate the coupling between motor variability and internal states, we devised a goal-directed reaching task that elicited movement variability both within and between human subjects. We first characterized this variability for our population of subjects during both planning and execution based on trial conditions. Then, to account for this variability, we built a simple behavioral model that incorporated dynamic internal states as accumulating trial history that evolve over time. Using computational methods, we used this model to explain differences in strategy across subjects by comparing their session performance to how they used internal states to inform future behavior. Finally, using recordings from intracranial depth electrodes implanted in the same subjects, we investigated the relationship between neural activity in large-scale brain networks and the encoding of these internal states.

### Motor task produced variability between and within subjects

Subjects performed a motor task using a robotic manipulandum for a virtual monetary reward using their dominant hand (Table 1). The task consisted of reaching movements towards a target at an instructed speed despite the possibility of physical perturbations (Fig. 1a). This task was designed to elicit movement variability both between- and within-subjects. We quantified this variability for the planning and execution phases of movement by calculating reaction time (RT) and speed error (SE), respectively. An overlay of RT and SE for each subject is shown in Fig. 1b, c. Group data comparisons confirmed the presence of between-subject variability in both RT and SE. That is, there was a main effect of subject on RT (three-way ANOVA: $F_{(9, 27)} = 7.24$, $p = 2.85 \times 10^{-3}$, partial $\eta^2 = 0.87$) and SE (three-way ANOVA: $F_{(9, 45)} = 13.17$, $p = 4.09 \times 10^{-12}$, partial $\eta^2 = 0.62$). We then used standard deviations (STD) to quantify the within-subject variability, where the absence of variability would mean STD equal to 0. Indeed, within-subject variability was consistently found across all subjects, meaning no subject was able to exactly reproduce the same movement across trials. All subjects had a non-zero STD for RT and SE, with subject 6 having the highest variability for RT and subject 2 having the highest

**Table 1 | Subject handedness and clinically relevant information such as identified epileptogenic zone**

| Subject | Handedness | Age at surgery (years) | Epileptogenic zone |
|---|---|---|---|
| 1 | right | 41 | Right hippocampus, Right entorhinal cortex, Right temporal pole |
| 2 | right | 34 | Right middle temporal gyrus, Right temporal pole, Right superior temporal sulcus |
| 3 | left | 37 | Left hippocampus |
| 4 | right | 36 | Right intraparietal sulcus, Right precuneus |
| 5 | right | 32 | Left insula (inconclusive) |
| 6 | left | 29 | Right hippocampus, Left superior temporal gyrus, Left orbitofrontal cortex |
| 7 | left | 23 | Left intraparietal sulcus, Left precuneus, Left supramarginal gyrus, Left angular gyrus, Left superior temporal gyrus |
| 8 | left | 26 | Left fusiform gyrus, Left hippocampus |
| 9 | right | 60 | Left temporal pole |
| 10 | right | 24 | Right hippocampus, Right fusiform gyrus |

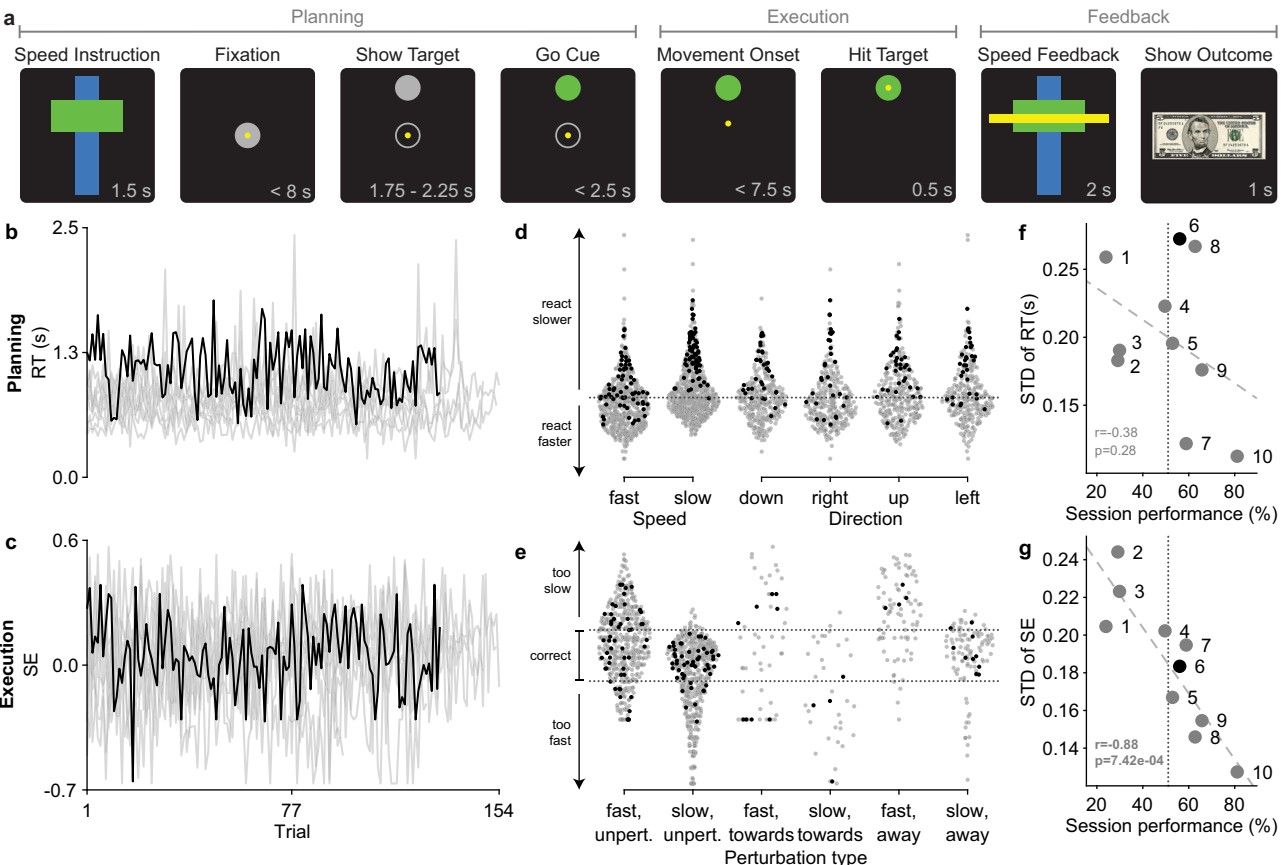

**Fig. 1 | Movement variability across subjects and trial conditions.** Summary of behavior during motor task shows variability within and between subjects that is independent of trial conditions yet correlates with their session performance. **a** A detailed timeline of epochs shown to subjects on a computer screen during an example trial. The name of each epoch is labeled above and the time each epoch was presented is displayed in the bottom right corner. The conditions (speed, direction, perturbation) for this example of a correct trial are fast, up, and unperturbed. Epochs were grouped into movement phases (planning, execution, feedback). Time-series of the observed (**b**) reaction time (RT) and (**c**) speed error (SE) across all trials and subjects. Subject 6 is colored black and the remainder of the subjects are colored gray. Observed (**d**) RT and (**e**) SE for all trials and subjects separated by trial conditions. Each marker represents the behavior of a subject during a trial for the specified condition. The arrows indicate the interpretation of

the behavior relative to the average. Subject 6 is colored black and the remainder of the subjects are colored gray. The gray dotted line in (**d**) indicates the average population RT (0.80 s). The gray dotted lines in (**e**) indicate the tolerance of SE between -0.13 and 0.13, where markers between these lines represent correct trials. Comparison of the variability of the observed (**f**) RT and (**g**) SE and performance across subjects. Standard deviation (STD) was used to quantify variability. Each marker is labeled by the subject it represents. Subject 6 is colored black and the remainder of the subjects are colored gray. The least-squares line is marked as the gray dashed line. Average session performance (51%) is marked by the vertical gray dotted line. There is a correlation between SE variability and session performance (two-tailed Pearson correlation: $r = -0.88$, $p = 7.42 \times 10^{-4}$), in which better performers have fewer variable errors, but not between RT variability and session performance. Source data are provided as a Source Data file.

variability for SE (Supplementary Tables 1–2). Combined, our motor task successfully produced behavior that varied between- and within-subjects.

We then investigated whether differences in trial conditions could explain within-subject variability. Figure 1d, e shows the distributions of the RT and SE for the population separated based on the trial condition. Starting with the planning phase, the trial conditions that influenced RT were speed and direction. We expected subjects to change their RT based on these speeds. Indeed, there was a main effect of speed on RT (three-way ANOVA: $F_{(1, 9)} = 9.74$, $p = 0.012$, partial $\eta^2 = 0.52$), meaning subjects reacted more quickly for fast trials than slow trials. We also found a main effect between the location of the target and how quickly subjects reacted (three-way ANOVA: $F_{(3, 27)} = 4.59$, $p = 0.0012$, partial $\eta^2 = 0.32$). Specifically, they reacted more slowly when the target was up compared to when the target was down (post-hoc Tukey's: $p = 0.0005$) or right (post-hoc Tukey's: $p = 0.0022$). However, we did not find a significant interaction between speed and direction. For the execution phase, the trial conditions that influenced SE were speed and perturbation. As a group, we found a main effect between the type of perturbation and SE (three-way

ANOVA: $F_{(5, 45)} = 23.23$, $p = 9.56 \times 10^{-12}$, partial $\eta^2 = 0.71$), with the exception of unperturbed compared to towards trials regardless of speed (Supplementary Table 10). However, we did find that RT does not significantly influence SE. Taken together, these results support a model of planning and execution using RT and SE that is both subject-specific and based on trial conditions. Supplementary Table 3 contains the complete summary of the trial conditions each subject experienced. See Supplementary Table 4–10 for the complete statistical test results.

We also found that performance differentiates variability between subjects. As previously described, variability was quantified using the STD of each behavior, where a higher STD corresponds to higher variable behavior. Session performance of each subject was quantified as the percent of correct trials over all completed trials. Average session performance was 51%. Figure 1f, g shows how subject's variability of RT and SE is related to their performance. Specifically, we found that subjects with higher session performance had less variable SE (two-tailed Pearson correlation: $r = -0.88$, $p = 7.42 \times 10^{-4}$).

In summary, even though all the subjects encountered the same trial conditions, their behavior varied which, in turn, affected their

performance. Therefore, factors other than trial conditions must be influencing their performance.

### Internal states capture movement variability

Using computational methods, we then developed a model to account for the variability that we observed between subjects (see Methods). Specifically, to account for variability not captured by trial conditions, we added two internal states. The first internal state was the error state, which accumulates the speed errors during past trials to keep track of how well a subject was accomplishing the instructed speed. The second internal state was the perturbed state, which accumulates the presence of perturbations during past trials to convey environmental uncertainty

To combine trial conditions and internal states, we used the state-space model illustrated in Fig. 2a. Each behavior (Eqs. (1) and (2)) was

calculated as a linear combination of the trial conditions and internal states (Eqs. (3) and (4)). These equations were then used to simultaneously estimate the behavior and internal states for all subjects. We were interested in examining (i) if our estimates follow the observed behavior, (ii) the characteristics that internal states and trial conditions independently capture, and (iii) how their internal states uniquely evolve to impact behavior. The model results of all subjects are shown in Supplementary Figs. 1–5. All model weights are in Supplementary Table 11.

Overall, we first found that the estimated behavior follows the observed behavior. We found a significant positive correlation between the estimated and observed behavior in 9 out of 10 subjects for RT (range of two-tailed Pearson correlation values: 0.3–0.78, Supplementary Fig. 1) and in 10 out of 10 subjects for SE (range of two-tailed Pearson correlation values: 0.42–0.86, Supplementary Fig. 2).

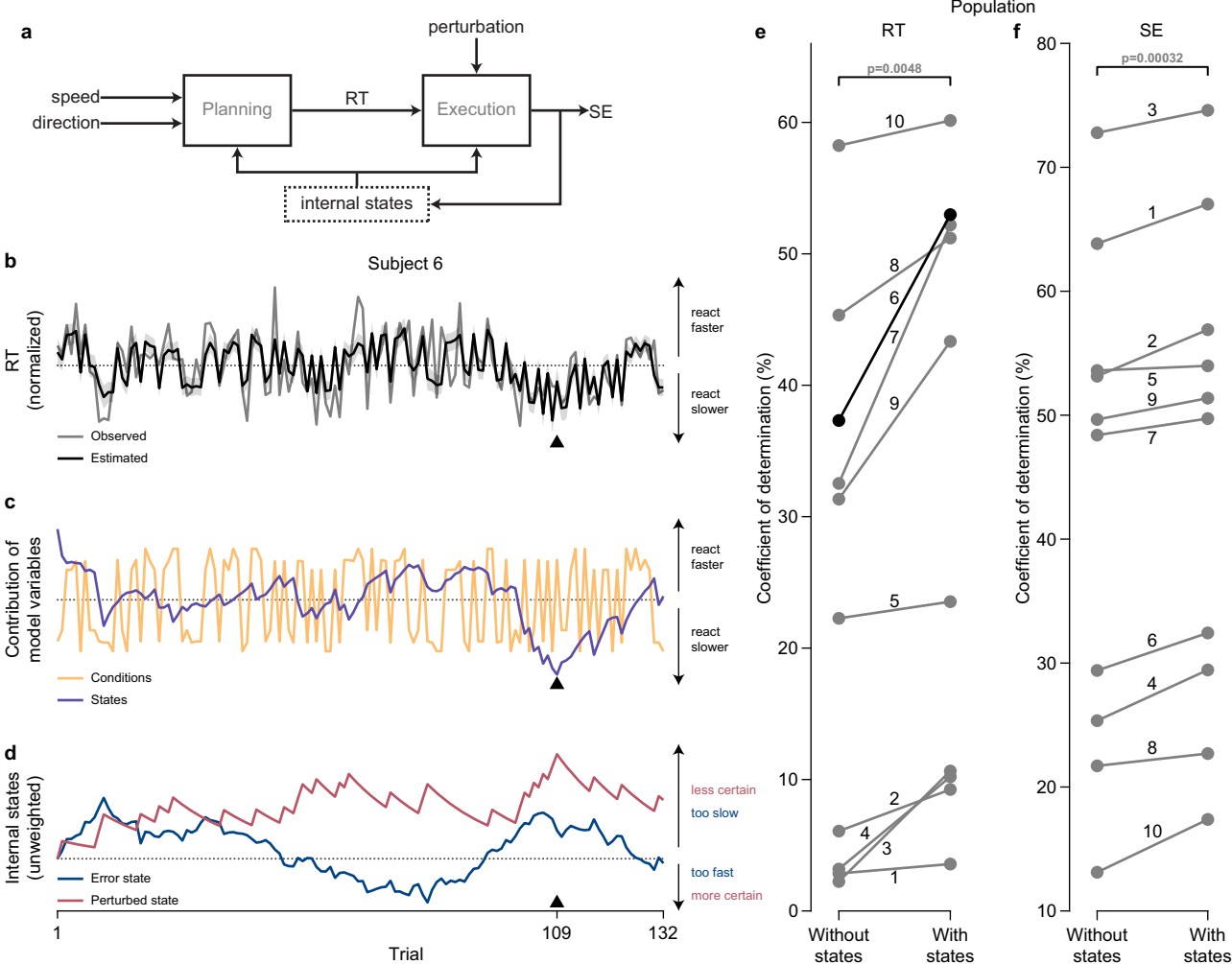

**Fig. 2 | Influence of model variables for estimating behavioral model. a** Block diagram of our dynamical model—representing the brain—that models behavior to capture movement variability. Internal states are outlined by the dotted line to highlight its latent feedback structure in the model. **b–d** Examination of model variables for subject 6 reveals underlying dynamics from the internal states that lead to the improvement of estimated behavior from the model observed across all subjects. The black triangle marks trial 109. **b** Time-series of the observed (gray solid line) and estimated (black solid line) reaction time (RT) ± the 95% confidence interval (gray shaded area) over all trials for subject 6. The gray dotted line markers their average RT (1.07 s). **c** Time-series of the conditions (orange solid line) and states (purple solid line) over all trials for subject 6. The gray dotted line markers their average RT. Adding the conditions and states together yields the estimated RT in (**b**). **d** Time-series of the error state (blue solid line) and perturbed state (pink

solid line) over all trials for subject 6. For demonstrative purposes, the states are not weighted but are scaled by their standard deviation. The gray dotted line marks 0. Adding the weighted combination of error and perturbed states together yields the states in (**c**). Goodness-of-fit for the (**e**) RT and (**f**) speed error (SE) models across all subjects measured using the coefficient of determination between the observed and estimated behavior, which ranges between 0% (worst) and 100% (best). We compared the behavioral models with internal states (With states) to a model with the same trial conditions but without internal states (Without states). Each marker is labeled by the subject it represents. Subject 6 is outlined in black. Adding internal states significantly improved the goodness-of-fit of the RT (two-tailed paired-sample $t$-test: $t(9) = -3.71$, $p = 0.0048$) and SE (two-tailed paired-sample $t$-test: $t(9) = -5.62$, $p = 0.00032$) model for all subjects, as larger values are better. Source data are provided as a Source Data file.

Only subject 1's RT was not statically significant (two-tailed Pearson correlation: $r = 0.19$, $p = 0.065$). To illustrate the relationship between the estimated and observed behavior, for example, consider subject 6, whose session performance was around the population average. Figure 2b shows subject 6's RT across all their trials, with their observed behavior in gray and their estimated behavior in black. Indeed, our estimates followed key features from their observed behavior (two-tailed Pearson correlation: $r = 0.73$, $p = 4.75 \times 10^{-23}$), including trial-to-trial changes and gradual changes such as between trials 100 and 125. For example, the estimate on trial 109 (black triangle) matches what was observed; subject 6 reacted faster than their average.

Next, we explored which parts of our model allowed the estimates to follow the observed behavior. Within the model, trial conditions accounted for the trial-by-trial changes and internal states accounted for gradual changes across all subjects (Supplementary Fig. 6a, b) for both RT (Supplementary Fig. 3) and SE (Supplementary Fig. 4). As such, the states captured the accumulation of subtle changes each subject exhibited. For example, the states (purple) for RT monotonically increased over trials for subjects 4, 5, 7, and 10 (Supplementary Fig. 3). This feature reproduced subjects progressively reacting slower possibly due to fatigue (Supplementary Fig. 1), which could not be replicated by the portion of the model responsible for trial conditions (orange). As another example, Fig. 2c shows the estimated RT of subject 6 separated into the trial conditions (orange) and internal states (purple). On trial 109 (black triangle) with the conditions slow and up, subject 6 should have reacted slower than average. However, the states outweighed the conditions, as demonstrated in subject 6 by their faster reaction time. By incorporating both trial conditions and internal states, our model captured features essential in realistic behavior that neither would be able to convey independently.

To determine why subjects behaved the way they did despite trial conditions, we next investigated the structure of the two internal states to understand their dynamics. The monotonic nature of the states observed in the population was carried over by one or both states for most subjects (Supplementary Fig. 5). We found the perturbed state (pink) to be responsible for the monotonic structure of the RT for subjects 4, 5, 7, and 10. This result suggests that these subjects reacted slower out of hesitancy originating from the accumulation of perturbations, making them less certain about their environment. Even subjects with nonmonotonic states exhibited trials with uncertainty. For example, Fig. 2d shows error state (blue) and perturbed state (pink) across all trials for subject 6. Their error and perturbed state on trial 109 (black triangle) are both positive, indicating that they recently moved slower than instructed and were perturbed. The perturbed state conveyed that recent successions of perturbations caused subject 6 to react and move faster in an attempt to counteract the uncertain environment. Overall, this data suggests that the internal states grant access to latent information about subjects.

Finally, we looked at our population of subjects to test whether adding internal states improved our ability to explain movement variability over using trial conditions alone, by comparing the coefficient of determination—a goodness-of-fit metric that measures the proportion of the behavioral variability that can be explained by the model variables—of the model with states to one without states. Figure 2e, f shows that adding the internal states to the model significantly improved the estimation two-tailed paired-sample $t$-test: $t(9) = -3.71$, $p = 0.0048$of both RT (two-tailed paired-sample $t$-test: $t(9) = -3.71$, $p = 0.0048$) and SE (two-tailed paired-sample $t$-test: $t(9) = -5.62$, $p = 0.00032$) across all subjects. The higher percentage means more of the variability is accounted for by the model variables. For example, the goodness-of-fit for subject 6 (black) improved by 16%. However, the model structure fits some subjects better than others. Specifically, subjects 1 through 4 had outlying model performance for RT (Fig. 2e)

compared to other subjects. This is notable because these subjects also had the lowest session performance. Since their model performance was low prior to adding internal states, it indicates that their RT varied by factors other than speed and direction as well as could explain their low session performance. Despite this, adding internal states also improved the deviance (Supplementary Fig. 6c, d) and 10-fold cross-validation using the fitted internal states for RT (Supplementary Fig. 6e). This, in addition to the fact that adding internal states improved their model fit and the inlying goodness-of-fit of their SE model, still merits the validity of interpreting their RT model.

In summary, internal states are essential for completely capturing movement variability. They conveyed slow evolving characteristics in the behavior, caused by retained trial history, that were not accounted for by the trial conditions.

## Learning from previous trials improves performance

The quantity of the weights on model variables reveals what each subject prioritized when they varied their behavior. Thus, we explored whether we could use our model to uncover different strategies used by subjects. Specifically, we hypothesized that subjects with higher session performance learned selective information from previous trials. Indeed, we found that they learned from the error state but not the perturbed state. We tested this by considering the relationship between subjects' weights on the internal states and session performance as a population (see Supplementary Fig. 7 all for model variables).

We found a negative correlation between the weights on the error state and session performance for both RT in Fig. 3a (two-tailed Pearson correlation: $r = -0.63$, $p = 0.05$) and SE in Fig. 3b (two-tailed Pearson correlation: $r = -0.74$, $p = 0.01$). Hence, higher performance corresponded to negative weights on the error state whereas lower performance corresponded to positive weights. Positive weights would cause high performing subjects to reduce their RT and SE after accumulating positive errors by moving slower than instructed. Under the same circumstances, lower performing subjects would continue to react and move more slowly than instructed due to the positive weights on the error state, thereby accumulating more errors. These results indicate that subjects with high performance adjusted their behavior to improve outcomes while subjects with low performance maintained their error tendencies. This finding suggests that higher performers learn based on feedback.

Alternatively, we did not foresee subjects learning from perturbed trials to improve their session performance due to the unpredictability of these trials. Indeed, Fig. 3c, d show that there was no relationship between the weight of the perturbed state and session performance on RT and SE, respectively. Instead, we found that subjects responded by either hesitating (i.e., react slower or move too slow) or counteracting (i.e., react faster or move too fast) for subsequent trials when they perceived the environment to be uncertain. More than half of the subjects positively weighted the perturbed state on RT, indicating hesitation to react. In terms of SE, we found half of the subjects moved slower (positive weight) in response to perturbation history while the other half moved faster (negative weight). We suggest that those who moved faster did so because they were exerting more force in their future movement in case they were perturbed.

In summary, our model could account for differences in the strategies of related to session performance, where higher performance corresponded to learning to counteract errors directly based on previous feedback. Though we could not find a complementary strategy against perturbations, we did find that subjects either hesitated or hastened to react when they perceived the environment to be uncertain. Another key strategy was speed, where larger magnitudes of weights on speed correlated with high session performance (Supplementary Fig. 7c).

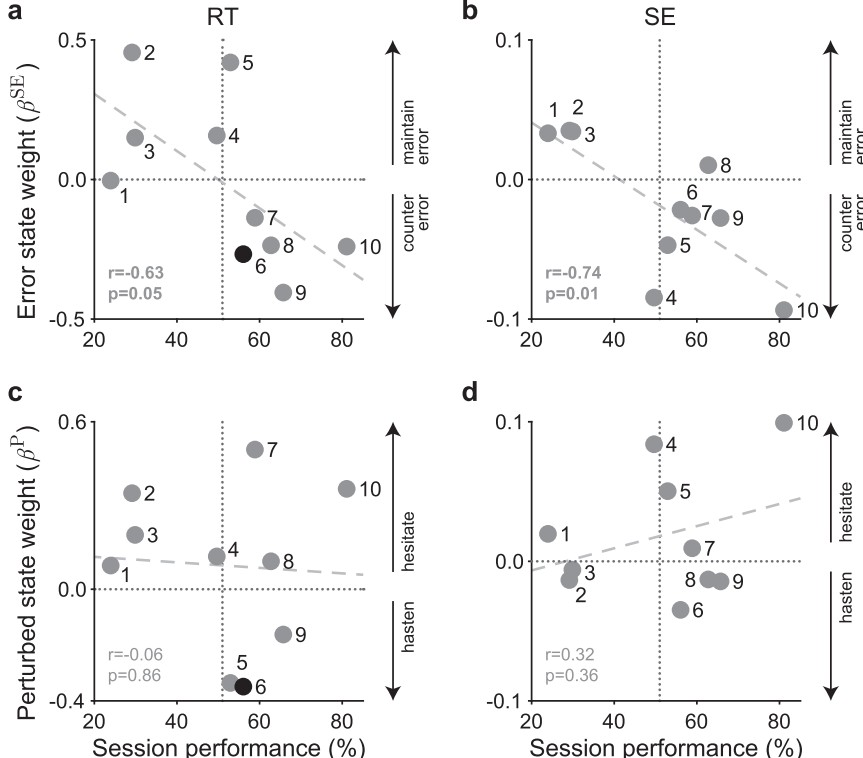

**Fig. 3 | Weights on internal states show subjects use different motor strategies based on session performance.** A comparison between the internal state weights and session performance reveals that higher performers learn from error and become more vigilant after perturbations. The larger the magnitude of the weight, the larger the impact the internal state has on behavior. The sign of the weight determines the impact the internal state has on behavior. Each marker is labeled by the subject. Average session performance (51%) is marked by the vertical gray dotted line. A weight of 0 (i.e., internal state does not impact behavior) is marked by the horizontal gray dotted line. The least-squares line is marked as the gray dashed line. All relationships were quantified using a two-tailed Pearson correlation without adjusting for multiple comparisons. **a** There is a significant relationship (two-tailed Pearson correlation: $r = -0.63$, $p = 0.05$) between session performance and weight of error state on RT. Higher performers countered their error by reacting

faster than average ($0 > \beta^{SE}$) and lower performers maintained their error by reacting slower than average ($0 < \beta^{SE}$) after moving too slow. **b** There is a significant relationship (two-tailed Pearson correlation: $r = -0.74$, $p = 0.01$) between session performance and weight of error state on speed error (SE). Higher performers tended to move too fast ($0 > \beta^{SE}$) and lower performers tended to move too slow ($0 < \beta^{SE}$) after moving too slow. **c** There is not a significant relationship between session performance and weight of perturbed state on RT. Most subjects hesitated after perturbation trials by reacting slower ($0 < \beta^{P}$). **d** There is not a significant relationship between session performance and weight of perturbed state on SE. Half of the subjects hesitated by moving slower ($0 < \beta^{P}$) and the other half hastened by moving faster ($0 > \beta^{P}$) after perturbations trials. Refer to Supplementary Fig. 6 and Supplementary Table 11 for all the weights. Source data are provided as a Source Data file.

## Large-scale brain networks encode internal states

Our combined experimental and modeling results provide support for the proposal that internal states can account for behavioral variability, within- and between-subjects. We next asked whether it was possible to gain insight into neural correlates of these internal states. More specifically, we asked whether such states are encoded by large-scale brain networks. To do this, we first assessed whether we could identify a set of brain regions linked to each internal state, and then determined which regions preferentially map to distinct large-scale brain networks related to session performance.

As a result of our unique experimental procedure, we had access to neural recordings from intracranial depth electrodes from each human subject simultaneously as they performed the motor task that was used to derive their internal states. Subjects were implanted with electrodes by clinicians to localize the epileptogenic zone for treatment. Illustrated in Fig. 4a, this granted us access to local field potential activity from a broad coverage of nonmotor regions, where we hypothesized the brain encoded internal states. These regions were first labeled anatomically using semi-automated electrode localization before being mapped to large-scale brain networks (see Methods, Supplementary Fig. 8a, b).

To identify the neural correlates amongst these regions with an unsupervised and data-driven method, we used a non-parametric cluster statistic[36] (see Methods) between the spectral data of each

region and each internal state across the population (Fig. 4b, c). This method finds windows of time (relative to epoch onset) and frequency (between 1 and 200 Hz) in which the spectral power of a region significantly correlates with the internal state across trials as demonstrated in Fig. 4d. The statistics provided us with two sets of neural correlates, one for each internal state.

Our modeling results showed that subjects weighed internal states differently, primarily based on their session performance. We suspected that this would be reflected in the brain for each subject by how well these regions encoded the states through the strength of their neural correlates. The degree to which a subject encodes an internal state in a region, which we called the encoding strength, was quantified by correlating the average power within the time-frequency window (from the population statistic) to the state on a trial-by-trial basis (see Methods). Figure 4e shows an example of how the encoding strength is obtained for a channel in subject 6 using the neural correlate from Fig. 4b–d.

## Dorsal attention network encodes error state

First, we found that the error state was encoded primarily by regions in the DAN (see Table 2 for details and Fig. 5a for visualization). The regions in DAN (dark blue) included the right intraparietal sulcus (IPS R), right middle temporal gyrus (MTG R), left superior frontal gyrus (SFG L), left superior parietal lobule (SPL L), and right superior

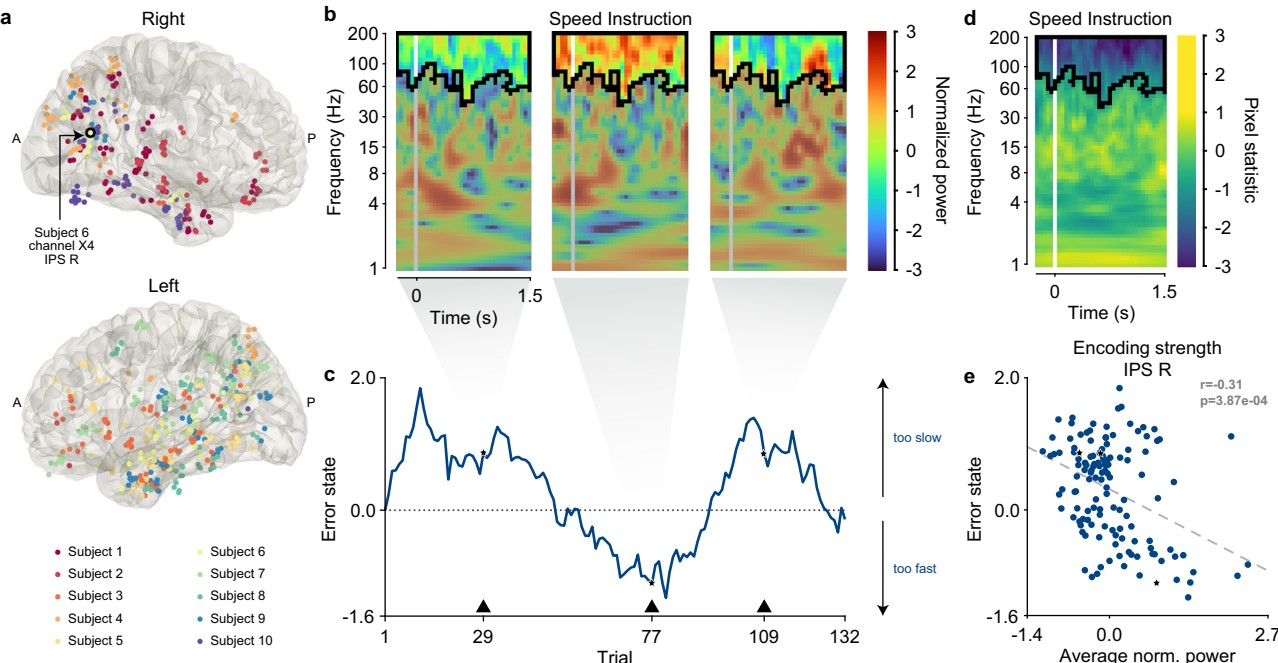

**Fig. 4 | Identifying correlates between neural data and internal states.**
**a** Subjects were implanted with multiple intracranial depth electrodes. The electrode coverage is visualized by mapping subject-specific coordinates of each channel onto a template brain called `cvs_avg35_inMNI152` from Freesurfer[94]. Electrodes are made up of multiple channels. For example, the channel clinically called X4 for subject 6, located in the right intraparietal sulcus (IPS R), is outlined in black where "A" means anterior and "P" means posterior. **b** The spectrogram of channel X4 in IPS R for subject 6 for three trials during Speed Instruction, time-locked to its onset (white solid line). It is represented by the time−frequency domain. The color of each pixel represents the normalized power. **c** Time-series of the error state (blue solid line) for subject 6. The black upward-pointing triangle marks the value of the error state for the trials corresponding to the spectrograms in (**b**). The non-parametric cluster statistic identified a significant cluster (outlined by the solid black line in (**b**)) where power correlated with the error state across all channels in the IPS R in the population. For example, the power in the cluster

increases for large positive magnitudes of the error state (too slow) in trials 29 and 109 but decreases when the error state is large and negative (too fast) in trial 77. **d** The statistical map over the time-frequency domain from the non-parametric cluster statistic between the IPS R and the error state across the population during Speed Instruction. Each pixel represents the hierarchical average across channels and subjects of the two-tailed $t$-statistic of the two-tailed Spearman's correlation between the error state and spectrogram. The white solid line markers where data was time-locked to epoch onset. Clusters were found as adjacent time-frequency windows that show a significant correlation (outlined in black). **e** There is a significant relationship (two-tailed Spearman's correlation: $r = -0.31$, $p = 3.87 \times 10^{-4}$) between the average normalized power in the cluster and the error state for the example in (**b**). This correlation coefficient also represents the encoding strength of this channel. The black star markers correspond to the trials in (**c**). The least-squares line is marked as the gray dashed line. Source data are provided as a Source Data file.

temporal sulcus (STS R). The second most prominent network was the visual network (yellow), which included the right parieto-occipital sulcus (POS R), left anterior transverse collateral sulcus (ATCS L), right cuneus (Cu R), and right inferior temporal gyrus (ITG R). Some regions from other networks also appeared, such as the right angular gyrus (AG R) and right posterior-dorsal cingulate gyrus (dPCC R) in DN as well as the right supramarginal gyrus (SMG R) from the Ventral Attention Network (VAN). However, in the regions we recorded from, the majority of the regions that encoded the error state were located in DAN. Seven of the 15 grouped clusters found to encode the error state belonged to the DAN. We verified the primacy of DAN encoding the error state over DN by comparing the distribution of the magnitude of their cluster statistics, where higher values indicate stronger clusters. Indeed, we found DAN to have a higher average cluster statistic than DN (two-tailed two-sample $t$-test: $t(53) = 2.32$, $p = 0.024$; Supplementary Fig. 8c).

Two groups of regions emerged based on (i) when during the trial did their activity correlate with the error state and (ii) in what frequency bands (see Methods). One-half of these regions encoded the error state throughout the session as persistent activity−activity that continues across all movement phases−in the frequency band hyper gamma (100−200 Hz). This activity was negatively correlated with the error state, meaning these regions exhibited higher deactivation when a subject moved slower than instructed. The other half encoded the error state as phasic activity−activity isolated to either planning,

execution, or feedback−in frequency bands <15 Hz. This activity was positively correlated with the error state for most regions. Overall, these regions use both persistent and phasic activity to encode the error state.

Recall our result above showing that subjects used opposing strategies regarding how they used their error state to change how they reacted in future trials based on their session performance (Fig. 3a, b). We next investigated whether this relationship would be reflected by how strongly these regions encodes the state. Since higher performance corresponded to subjects learning from their errors, we expected a positive correlation between encoding strength and session performance. We found this to be true (two-tailed Pearson correlation: $r = 0.66$, $p = 1.80 \times 10^{-8}$), as shown in Fig. 5b with DAN in dark blue, DN in red, VAN in light blue, and the visual network in yellow. For example, the SPL L−a key hub of DAN−modulates its neural activity based on the error state for a subject with above average performance (subject 9) but not for a subject with below average performance (subject 4) (Fig. 5c).

**Default and dorsal attention networks encode perturbed state**
Second, we further found that the perturbed state was encoded primarily by regions in the DN and DAN (see Table 3 for details and Fig. 5d for visualization). The regions in DN (red) included the right angular gyrus (AG R), left anterior cingulate gyrus (ACG L), right middle temporal gyrus (MTG R), right posterior-dorsal cingulate gyrus (dPCC R),

**Table 2 | List of significant clusters across brain regions and epochs encoding the error state**

| Brain region | Hemi. | Acr. | Freq. | Network | Subj. | Onset (%) | Planning — Speed Instruction | Planning — Fixation | Planning — Show Target | Go Cue | Execution — Movement Onset | Execution — Hit Target | Feedback — Speed Feedback | Feedback — Show Outcome |
|---|---|---|---|---|---|---|---|---|---|---|---|---|---|---|
| angular gyrus | right | AG R | hyper gamma | Default | 5 | 0 | $-9.99\times10^{-4}$ | $-9.99\times10^{-4}$ | $-9.99\times10^{-4}$ | $-9.99\times10^{-4}$ | $-9.99\times10^{-4}$ | $-9.99\times10^{-4}$ | $-9.99\times10^{-4}$<br>$-9.99\times10^{-4}$ | $-9.99\times10^{-4}$ |
| intraparietal sulcus | right | IPS R | hyper gamma | Dorsal attention | 4 | 0 | $-9.99\times10^{-4}$ | $-9.99\times10^{-4}$ | $-9.99\times10^{-4}$ | $-9.99\times10^{-4}$ | $-9.99\times10^{-4}$ | $-9.99\times10^{-4}$ | $-9.99\times10^{-4}$<br>$-9.99\times10^{-4}$ | $-9.99\times10^{-4}$ |
| superior parietal lobule | left | SPL L | hyper gamma | Dorsal attention | 3 | 8 | $-9.99\times10^{-4}$ | $-9.99\times10^{-4}$ | $-9.99\times10^{-4}$ | $-9.99\times10^{-4}$ | $-9.99\times10^{-4}$ | $-9.99\times10^{-4}$ | $-9.99\times10^{-4}$ | $-9.99\times10^{-4}$ |
| superior temporal sulcus | right | STS R | hyper gamma | Dorsal attention | 5 | 37 | $-9.99\times10^{-4}$ | $-9.99\times10^{-4}$ | $-9.99\times10^{-4}$ | $-9.99\times10^{-4}$ | $-9.99\times10^{-4}$ | $-9.99\times10^{-4}$ | $-9.99\times10^{-4}$ | $-8.99\times10^{-3}$<br>$-4.00\times10^{-3}$ |
| supramarginal gyrus | right | SMG R | hyper gamma | Ventral attention | 4 | 0 | $-9.99\times10^{-4}$ | $-9.99\times10^{-4}$ | $-9.99\times10^{-4}$ | $-9.99\times10^{-4}$ | $-9.99\times10^{-4}$ | $-9.99\times10^{-4}$ | $-9.99\times10^{-4}$ | $-9.99\times10^{-4}$ |
| transverse temporal sulcus | left | TTS L | hyper gamma | Default | 3 | 0 | $-9.99\times10^{-4}$<br>$-3.00\times10^{-3}$ | $-9.99\times10^{-4}$ | $-9.99\times10^{-4}$ | $-3.00\times10^{-3}$ | $-2.00\times10^{-3}$<br>$-2.00\times10^{-3}$ | $-9.99\times10^{-4}$ | $-9.99\times10^{-4}$<br>$-5.00\times10^{-3}$ | $-9.99\times10^{-4}$ |
| parieto-occipital sulcus | right | POS R | hyper gamma | Visual | 3 | 0 | $-9.99\times10^{-4}$ | | | | | | | |
| cuneus | right | Cu R | delta | Visual | 3 | 0 | | | $9.99\times10^{-4}$ | | | | | |
| posterior-dorsal cingulate gyrus | right | dPCC R | delta | Default | 3 | 0 | | | $5.00\times10^{-3}$ | | | | | |
| middle temporal gyrus | right | MTG R | hyper gamma | Dorsal attention | 6 | 4 | | | | $-3.00\times10^{-3}$ | $-9.99\times10^{-4}$ | $-3.00\times10^{-3}$ | | |
| superior frontal gyrus | left | SFG L | theta | Dorsal attention | 3 | 0 | | | | | $9.99\times10^{-4}$ | $2.00\times10^{-3}$ | | |
| cuneus | right | Cu R | delta | Visual | 3 | 0 | | | | | | | $9.99\times10^{-4}$ | |
| middle temporal gyrus | right | MTG R | delta | Dorsal attention | 6 | 4 | | | | | | | $2.00\times10^{-3}$ | |
| superior parietal lobule | left | SPL L | alpha | Dorsal attention | 3 | 8 | | | | | | | $9.99\times10^{-4}$ | |
| inferior temporal gyrus | right | ITG R | theta | Visual | 4 | 0 | | | | | | | | $3.00\times10^{-3}$ |

For each brain region and task epoch, the table reports all clusters that pass a false discovery rate of level q = 0.015 and their level within the cell. For each brain region, the table also provides the hemisphere (Hemi.), acronym (Acro.), dominant frequency band (Freq.), network, number of subjects with recordings in this region (Subj.), and percentage of electrodes in this region that have been annotated as part of an onset zone (Onset).

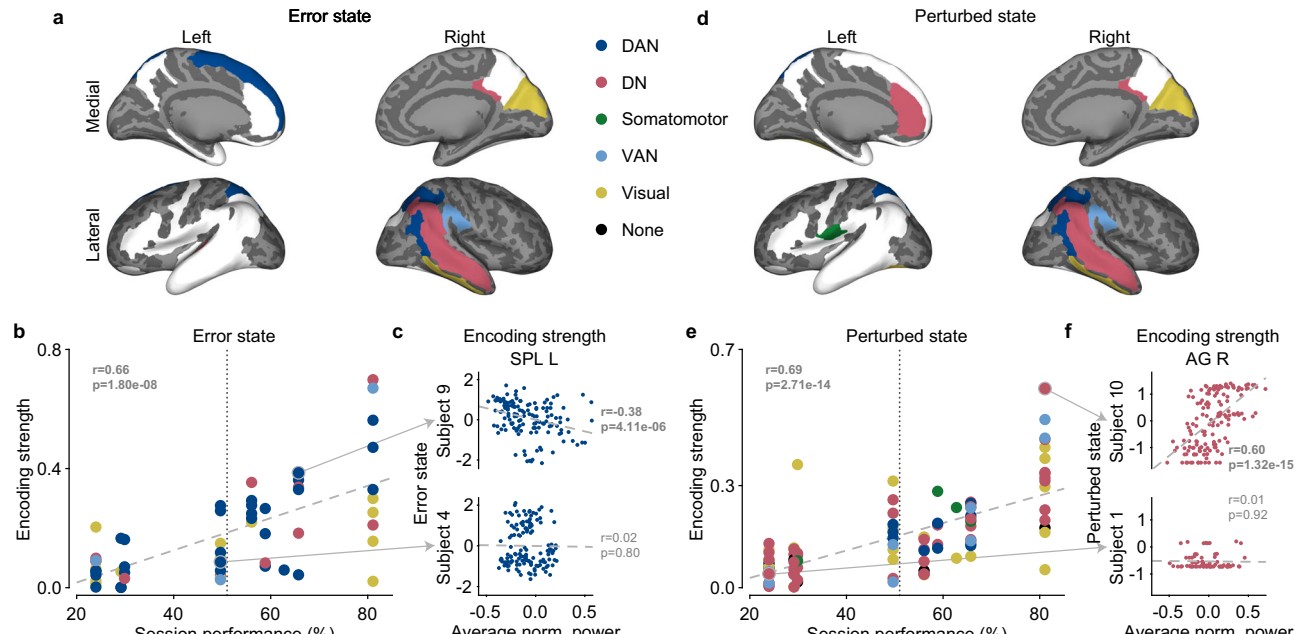

**Fig. 5 | Large-scale brain networks encode internal states and reflect session performance.** Summary of regions found by our analysis that encodes the (**a**) error state and (**d**) perturbed state. They are highlighted on an inflated template brain called `cvs_avg35_inMNI152` from Freesurfer[94] based on the large-scale brain network they belong to: dorsal attention (DAN) in dark blue, default (DN) in red, somatomotor in green, ventral attention (VAN) in light blue, visual in yellow, and none in black. The light gray represents the gyri and the dark gray represents the sulci. The white represents regions included in our analysis but were not found to encode the state based on performance. Subjects encoded the (**b**) error state (two-tailed Pearson correlation: $r = 0.66$, $p = 1.80 \times 10^{-8}$) and (**e**) perturbed state (two-tailed Pearson correlation: $r = 0.69$, $p = 2.71 \times 10^{-14}$) in their neural activity based on their session performance. Each marker represents the average encoding strength of a region for a subject and is colored by the network the region belongs to. An encoding strength of 1 means the state is strongly encoded by the activity of a region whereas 0 means the state is not encoded. Average session performance (51%) is marked by the vertical gray dotted line. The least-squares line is marked as the gray dashed line. Subjects with higher performance had neural activity that modulated significantly more with either internal states than subjects with lower performance. Examples of subjects with high and low session performance for the (**c**) error state and (**f**) perturbed state. The left superior parietal lobule (SPL L) between subject 9 and subject 4 in the DAN and the right angular gyrus (AG R) between subject 10 and subject 1. Each marker represents a trial, with the corresponding neural activity of the cluster for a channel in the region and normalized state value. The gray dashed line represents the least-square line. The two-tailed Spearman's correlation (**r**) and p-value (**p**) are included in each panel, where the correlation magnitude was used as the encoding strength. Source data are provided as a Source Data file.

and right superior temporal sulcus (STS R). The regions in DAN (blue) included IPS R and SPL L. As with the error state, regions in the visual network (yellow) also encoded the perturbed state. They consisted of the right cuneus (Cu R), right inferior temporal gyrus (ITG R), left fusiform gyrus (FuG L), and right parieto-occipital sulcus (POS R). Other networks that appeared included the VAN (SMG R) and even the somatomotor network through the left subcentral gyrus (SubCG L). The subcortical region right hippocampus (Hippo R) briefly encoded the perturbed state after execution. However, since we were only interested in large-scale brain networks of the neocortex, Hippo R was not classified in this study. Although the majority of regions we found to encode the perturbed state were in DN, we did not find a significant difference between the magnitude of the cluster statistics in DN and DAN (two-tailed two-sample t-test: $t(22) = 0.16$, $p = 0.87$; Supplementary Fig. 8d).

The regions in DN and DAN that encoded the perturbed state did so with phasic activity in frequency bands under 15 Hz (i.e., delta, theta, alpha). Most regions had activity that was positively correlated with the perturbed state; trials with a high perturbed state (associated with recent perturbations) coincided with higher activation in DN and DAN.

Recall our earlier result in which subjects alter their behavior (i.e., hesitate or hasten) with regard to the perturbed state. Although there was no consistent strategy based on session performance, we still speculated whether there would be a neurological relationship between session performance and how the perturbed state was encoded. As with the error state, we expected subjects with above average performance to have higher encoding strength because they

generally weighted the perturbed state with a higher magnitude than those with below average performance (Fig. 3c–d). Indeed, Fig. 5e shows the networks that modulate their neural activity with the perturbed state based on session performance (two-tailed Pearson correlation: $r = 0.69$, $p = 2.71 \times 10^{-14}$). This figure also shows regions from other networks, which include the DAN in dark blue, VAN in light blue, somatomotor network in green, and visual network in yellow. For example, as a hub for the DN, the AG R would increase activity when the perturbed state was high (i.e., after perturbation trials) for subjects with high session performance (subject 10) but for subjects with low session performance (subject 1) (Fig. 5f). Even though our models could not find a consistent strategy based on session performance, subjects with above-average performance still modulated their activity to match the perturbed state.

## Connectivity correlates with performance and strategy

Recall that the weight a subject puts on their internal states provides insight into their learning strategies. Since subjects encode the internal states in distinct networks, we hypothesized that these strategies would further be reflected by the functional connectivity of the networks that we identified as encoding both the internal state and session performance. Simply put, pairs of regions that are spatially separate yet whose neural activity is correlated are functionally connected[37]. We quantified functional connectivity by correlating the average power within the time-frequency window used for encoding strength on a trial-by-trial basis for each channel per subject (Fig. 6a, b). Taking the average of these correlations resulted in a value

**Table 3 | List of significant clusters across brain regions and epochs encoding the perturbed state**

| Brain region | Hemi. | Acr. | Freq. | Network | Subj. | Onset (%) | Planning | | | Execution | | | Feedback | |
|---|---|---|---|---|---|---|---|---|---|---|---|---|---|---|
| | | | | | | | Speed Instruction | Fixation | Show Target | Go Cue | Movement Onset | Hit Target | Speed Feedback | Show Outcome |
| inferior temporal gyrus | right | ITG R | delta | Visual | 4 | 0 | $1.10 \times 10^{-2}$ | | | | | | | |
| intraparietal sulcus | right | IPS R | theta | Dorsal attention | 4 | 0 | $2.00 \times 10^{-3}$ $9.00 \times 10^{-3}$ | | | | | | | |
| cuneus | right | Cu R | high gamma | Visual | 3 | 0 | $-9.99 \times 10^{-4}$ | $-7.99 \times 10^{-3}$ | | | | | | |
| superior parietal lobule | left | SPL L | alpha | Dorsal attention | 3 | 8 | $9.99 \times 10^{-4}$ | $6.99 \times 10^{-3}$ | | | | | | |
| superior temporal sulcus | right | STS R | theta | Default | 5 | 37 | $9.99 \times 10^{-4}$ | $9.99 \times 10^{-4}$ | | | | | | |
| supramarginal gyrus | right | SMG R | theta | Ventral attention | 4 | 0 | $9.99 \times 10^{-4}$ | $9.99 \times 10^{-4}$ | | | | | | |
| inferior temporal gyrus | right | ITG R | theta | Visual | 4 | 0 | $2.00 \times 10^{-3}$ | $9.99 \times 10^{-4}$ | | $2.00 \times 10^{-3}$ | | | | |
| angular gyrus | right | AG R | theta | Default | 5 | 0 | $9.99 \times 10^{-4}$ | $9.99 \times 10^{-4}$ | $9.99 \times 10^{-4}$ $9.99 \times 10^{-4}$ | $9.99 \times 10^{-4}$ | | | | |
| anterior cingulate gyrus | left | ACG L | high gamma | Default | 3 | 0 | | $-5.00 \times 10^{-3}$ | | | | | | |
| middle temporal gyrus | right | MTG R | alpha | Default | 6 | 4 | | $1.10 \times 10^{-2}$ | | | | | | |
| posterior-dorsal cingulate gyrus | right | dPCC R | theta | Default | 3 | 0 | | $6.99 \times 10^{-3}$ | | | | | | |
| posterior-dorsal cingulate gyrus | right | dPCC R | hyper gamma | Default | 3 | 0 | | $3.00 \times 10^{-3}$ | $4.00 \times 10^{-3}$ | | | | | |
| subcentral gyrus | left | SubCG L | hyper gamma | Somatomotor | 4 | 0 | | $2.00 \times 10^{-3}$ | $2.00 \times 10^{-3}$ | $9.99 \times 10^{-4}$ | $1.10 \times 10^{-2}$ | | $3.00 \times 10^{-3}$ | $2.00 \times 10^{-3}$ |
| parieto-occipital sulcus | right | POS R | delta | Visual | 3 | 0 | | | $9.99 \times 10^{-4}$ | | | | | |
| supramarginal gyrus | right | SMG R | theta | Ventral attention | 4 | 0 | | | $1.10 \times 10^{-2}$ | $2.00 \times 10^{-3}$ | $1.10 \times 10^{-2}$ | | | |
| superior parietal lobule | left | SPL L | delta | Dorsal attention | 3 | 8 | | | | $9.99 \times 10^{-4}$ | | | | |
| fusiform gyrus | left | FuG L | delta | Visual | 4 | 0 | | | | $5.00 \times 10^{-3}$ | $6.99 \times 10^{-3}$ | | | |
| anterior cingulate gyrus | left | ACG L | high gamma | Default | 3 | 0 | | | | | $-5.99 \times 10^{-3}$ | $-3.00 \times 10^{-3}$ | | |
| middle temporal gyrus | right | MTG R | alpha | Default | 6 | 4 | | | | | $9.99 \times 10^{-4}$ | $1.10 \times 10^{-2}$ | $6.99 \times 10^{-3}$ | |
| hippocampus | right | Hippo R | theta | None | 5 | 60 | | | | | | $7.99 \times 10^{-3}$ | $1.10 \times 10^{-2}$ | |
| inferior temporal gyrus | right | ITG R | alpha | Visual | 4 | 0 | | | | | | | $9.99 \times 10^{-4}$ | |
| superior temporal sulcus | right | STS R | delta | Default | 5 | 37 | | | | | | | $6.99 \times 10^{-3}$ | |
| superior temporal sulcus | right | STS R | theta | Default | 5 | 37 | | | | | | | | $1.20 \times 10^{-2}$ |

For each brain region and task epoch, the table reports all clusters that pass a false discovery rate of level q = 0.015 and their level within the cell. For each brain region, the table also provides the hemisphere (Hemi.), acronym (Acro.), dominant frequency band (Freq.), network, number of subjects with recordings in this region (Subj.), and the percentage of electrodes in this region that have been annotated as part of an onset zone (Onset).

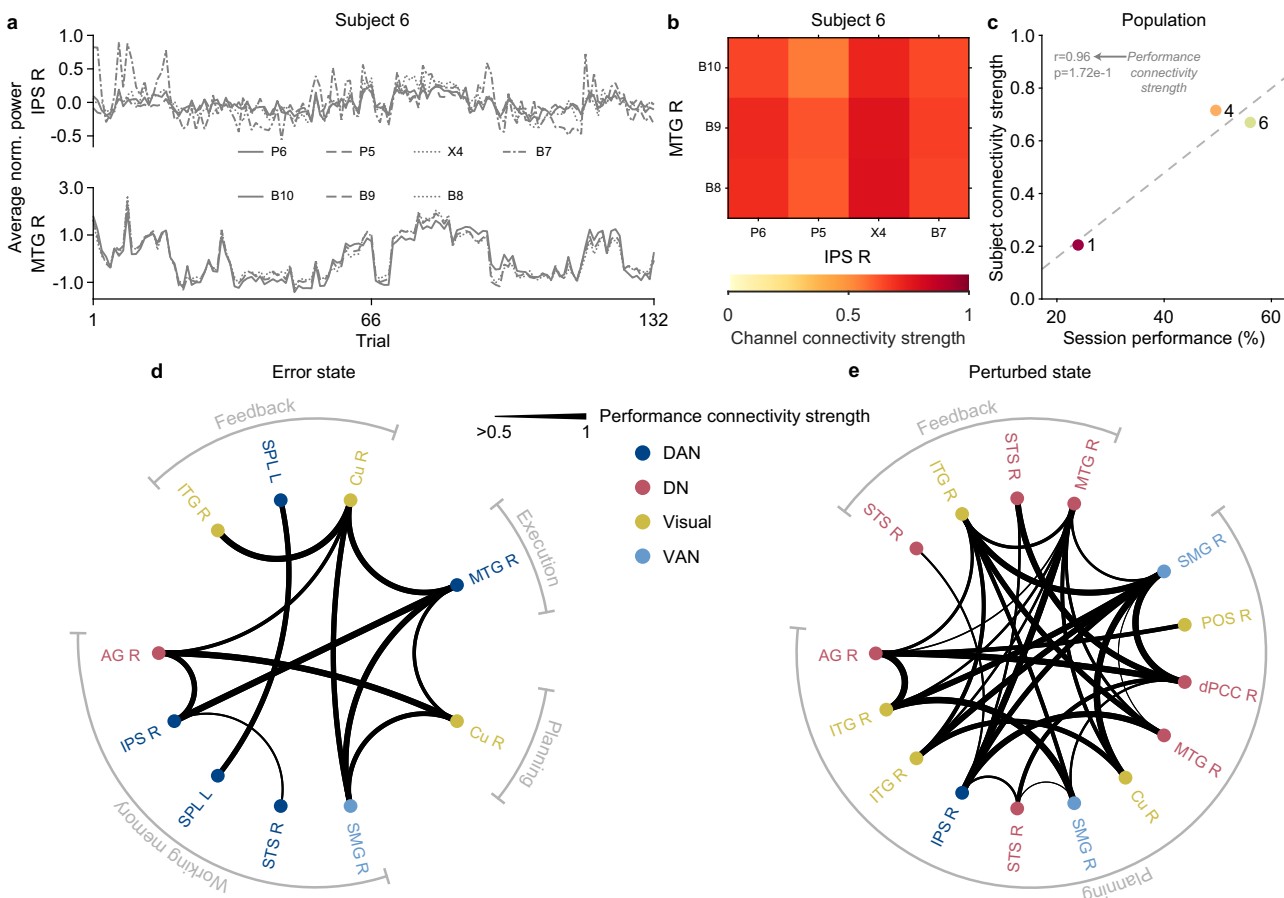

**Fig. 6 | Performance-based network connectivity supports learning strategy.**
**a** Average normalized power for channels in the right intraparietal sulcus (IPS R) and right middle temporal gyrus (MTG R) of subject 6 across trials. The value for each trial comes from averaging the normalized power in the time-frequency windows identified by the non-parametric cluster statistic (Table 2 and Table 3). This figure uses the results of the error state. **b** Matrix of channel connectivity strengths found by taking the magnitude of two-tailed Spearman's correlation value between the average normalized power of channels in IPS R and MTG R. Values close to 1 (red) indicate high correlations and values close to 0 (yellow) indicate low correlations. **c** Comparison between subject connectivity strength (found by taking the average of the channel connectivity strength) and session performance reveals high-performing subjects engaged in stronger connectivity between IPS R and MTG R (two-tailed Pearson correlation: $r = 0.96$, $p = 0.17$). The

correlation value represents the performance connectivity strength. Directed graph depicting performance connectivity strength across regions for (**d**) error state and (**e**) perturbed state. The regions are colored by the large-scale brain network they belong to: dorsal attention (DAN) in dark blue, default (DN) in red, ventral attention (VAN) in light blue, and visual in yellow. Only pairs with positive relationships between connectivity strength and session performance are shown (i.e., higher performance with higher subject connectivity). The magnitude of the correlation between connectivity strength and session performance is depicted by the thickness of the edge; the higher the correlation, the thicker the edge. Values below 0.5 were truncated. The regions are ordered by when they encode each state and its phase is labeled along the circumference, demonstrating synchrony across time. See Supplementary Fig. 9 for full performance connectivity. Source data are provided as a Source Data file.

that summarizes the functional connectivity between a pair of regions called subject connectivity strength. Population connectivity strength, found by averaging subject connectivity strength across all pairs, is summarized in Supplementary Fig. 9.

Unfortunately, we were not able to observe all possible pairs of regions because either the pair was not represented in the data set, or we did not have enough subjects with the pair ($n \geq 3$) to make any remarks. Based on behavioral results, we were interested in the variability of connectivity between subjects. Specifically, since connectivity strength is subject-specific, we expected it to vary based on how subjects performed related to different strategies we observed earlier. For example, subjects with high performance will exhibit higher subject connectivity strength between key regions in distinctive networks (Fig. 6c).

Our results above demonstrated that high session performance was accompanied by subjects compensating their behavior in response to the error state. Using the set of regions that encode the error state, Fig. 6d further shows that subjects with high performance

favored connectivity between regions in the DN (red), DAN (dark blue), VAN (light blue), and visual network (yellow). We found that regions that encoded with persistent activity connected to regions that encoded with phasic activity. Specifically, the persistent activity from DAN (IPS R and SPL L) and VAN (SMG R) projected namely to regions in the visual network during key phases during the trial (i.e., planning, execution, feedback). This result suggests that the error state is held and distributed by the attention networks to modulate visual attention. An increase in such attention could account for the counteracting behavior observed by top performers, such as reacting faster after moving slower than instructed (Fig. 3a). We also observed a significant correlation between the persistent activity and phasic activity during feedback for the SPL L, which is a key hub of DAN. This relationship implicates persistent and phasic activity having different roles when encoding the error state. Perhaps the phasic activity represents the various sensory processes (depending on the phase) used to extrapolate information about the error state and the persistent activity holds this information in memory for accessibility by other regions.

Then, the connection between SPL L could be an example of updating between integration and memory. Overall, subjects with high session performance favored connections between DAN and visual networks to encode the error state. These connections could account for how these subjects learned from their errors, specifically by updating their memory based on visual feedback to modulate attention in future trials.

These results also revealed that subjects with high session performance altered how they reacted (i.e., hesitate or hasten) in response to the perturbed state. Figure 6e further shows the connectivity between the perturbed state encoding regions favored towards higher session performance. Notable, we found more connections for the perturbed state than compared to the error state. This can be interpreted as the perturbed state being more disruptive to their behavior and requiring the integration of sensorimotor pathways for interpretation during planning and feedback. In general, most of the connections we found were between DN and visual network. This relationship could allow for the communication of visual information from feedback to be available for the DN during planning, such as to update their expectation about the environment of the motor task. This information could then be projected to regions in the visual network and VAN to plan their future behavior. For example, dPCC R (hub of DN) during early planning projects to SMG R (hub of VAN) during late planning and early execution. Taken together, these results suggest that the DN could be modulating bottom-up visual attention based on the perturbed state. There are also notable connections between DN and DAN: IPS R and dPCC during planning as well as IPS R from planning to MTG R during feedback. These junctions suggest points when DN and DAN communicate task-relevant information that could also aid in modulating visual attention. Overall, high session performance favored connectivity between DN to other relevant networks whose activity modulates based on the perturbed state during planning and feedback, which they could have used to adapt their responsiveness based on perturbations we observed from their behavior.

## Discussion

In the present study, we first identified two internal states—based on error and environmental history—that induce movement variability in humans. The degree to which states contribute to an individual's variability reveals different strategies based on session performance regarding how they used their states to inform future behavior. Remarkably, we then found that these internal states were linked to encoding in large-scale brain networks, DAN and DN. Taken together our findings reveal that differences in large-scale brain networks that can distinguish subjects by their session performance: (i) high performers modulate network activity on a trial-by-trial basis with respect to their internal states and (ii) their learning strategy is supported by explicit connections within and between networks during phases of movement.

The general effect of error and the environment on movement variability are well documented in motor control. Traditionally, the goal of optimal feedback control is to minimize error during movement, where larger errors require more variability to correct the movement[38,39]. By accounting for the accumulation of errors from trial-to-trial, we also found that variability scaled based on error history for those with above average session performance. In everyday life, we adapt our behavior to fit the environment based on prior experience. However, it is difficult to adapt when disturbances are rare and unpredictable. Fine and Thoroughman found that it is difficult for subjects to learn how to respond to these disturbances that occur for <20% of trials[40]. They proposed an adaptive switch strategy that depends on the environmental dynamics: ignore performance from trials with rare disturbances and learn when they are common[41]. Complimentary, we found that subjects responded in trials after

perturbations by either reacting hesitantly or vigorously. Nevertheless, their strategy was not a predictor of how well they performed our reaching task, which can be further explored in future studies.

To learn our speed-instructed motor task, subjects needed to keep track of errors using working memory. Some of their errors were self-inflicted (e.g., forgetting the speed instruction) while others were caused by unexpected perturbations. In either case, we found that our subjects learned the task by monitoring their history of errors to decide when to allocate attention using the DAN. Our findings are in agreement with our current knowledge about DAN, specifically for its control of visuospatial attention[42]. Other studies have found that it activates before and during expected (top-down) as well as after unexpected (bottom-up) visual stimuli[43]. Therefore, DAN appears to combine bottom-up information (i.e., unexpected perturbations) with top-down information (i.e., self-inflicted errors) when deciding how much attention to allocate for future trials.

More specifically, we found DAN encoded using two frequency patterns: activating below 15 Hz and deactivating in frequencies above 100 Hz. The former was found when subjects either recently moved faster than instructed—to which they responded by slowing down—or after recently perturbed trials. This finding is reminiscent of the speed-accuracy trade-off phenomenon. Speed-accuracy trade-off is observed during motor learning as a form of behavioral variability where subjects must balance moving faster at the cost of making more errors to optimize performance[44]. This phenomenon innately requires tracking history[45]. This suggests that DAN is not only tracking history but is modulating it for learning. Other fMRI studies also corroborate our observation. Increases in Blood-Oxygen-Level-Dependent (BOLD) signal relative to baseline in DAN have been reported when subjects were instructed to prioritize speed (instructed as fast or slow) over accuracy during a response interference speed-accuracy trade-off task[46]. Another study using an anti-saccade task found DAN activation through BOLD signal positively correlated with RT (i.e., more activation when slowing down) and being the least activated on trials with large errors (which compares to our task when the error state would be close to zero)[47]. In a visually guided motor sequence learning task, DAN activated—through BOLD signal—to large errors during early learning, which they related to active visuospatial attention when first learning the sequence[48]. Taken together, our results directly implicate DAN as a network for encoding tracking history.

The persistent activity of DAN in frequencies above 100 Hz when found are characteristic of working memory[49–51]. During motor planning in tasks with working memory, DAN has been shown to maintain task-relevant information, such as target location, during delay periods using persistent activity in both whole-brain and single unit recordings[52,53]. DAN has also been linked to working memory closely tied to visual attenuation related to memory load and top-down memory attention control during visual working memory tasks[54]. Though our task does not explicitly study working memory, given the evidence, our results suggest that DAN is tracking the accumulation of past errors in working memory. This would also support our connectivity results as information held in working memory can be easily accessed by multiple systems, such as for sensorimotor integration or visual processing, for recalling and updating[55–57].

Finally, we found that the encoding strength of regions in the DAN and functional connectivity between these regions, along with the visual network, scaled based on the subject's overall performance. Though these results are preliminary, we linked this with observations that subjects have different strategies. Those with above average session performance more strongly encoded error history in and between the regions in the DAN. Hence, they were more engaged in the task and modulated their attention based on the error state. This led to them slowing down after they moved too quickly and vice versa as predicted by their model weights. Meanwhile, below average performers have poor attentional control and memory capacity and thus did not learn

from their mistakes. Studies have shown that poor overall behavioral performance is related to decreased activity in DAN[58–62] (called out of the zone[63]), fluctuations in attention and working memory known as a lapse in attention[64–66], and poor connectivity in DAN[67,68]. Clinically, studies of Attention Deficit Hyperactivity Disorder (ADHD) found compromised performance during working memory tasks is related to poor attentional control in DAN-related regions[69,70], similar to what we observed in our poor performers.

The random perturbations made our task more difficult by creating uncertainty in the environment. Our models show that subjects also kept track of past perturbations suggesting possible attempts to learn the environment. We found that regions whose activity correlated with the perturbed state were in DN and DAN. They did so primarily in the frequency bands theta and alpha (<15 Hz). The involvement of DAN and the perturbed state was discussed with the error state for its connection to allocating attention. As for DN, it encoded the perturbed state by increasing activity when the environment was perceived to be more uncertain. This function of the DN is similar to a recent study by Brandman et al.[71] in which they found that the DN activated immediately following unexpected stimuli in the form of surprising events during movie clips. They suggested that the DN could be involved in prediction-error representation, which our results also support. Furthermore, they also found that DN coactivated with the hippocampus during unexpected stimuli. This finding parallels a previous report from our dataset which demonstrated activation of the hippocampus in response to motor uncertainty[72]. A proposed process model of reinforcement learning incorporates regions in the DN and hippocampus that predict and evaluate the semantic knowledge about the environment to inform future behavior[73]. Taken together, our findings suggest that we captured the DN responding to the unexpected stimuli by updating semantic knowledge about the environment which informs future behavior based on the perturbed state.

Behaviorally, our model did not establish a link between subjects' performance and how they handled past perturbations but our analysis of neural activity revealed that subjects with high session performance demonstrated increased activation of regions in DN in response to the history of perturbations as well as correlated activity across regions between trials. Since subjects have no control over the perturbations, we speculate that they attempted to learn how to react in a way that optimizes their performance. Although they are applied to random trials and in random directions, perturbations were only applied during the beginning of the movement. Therefore, subjects can learn to prepare themselves in a way they see fit. Our preliminary findings suggest that high performers effectively implement their new semantic knowledge about the environment to explore different approaches to prepare for the possibility of future perturbations. DN becomes more activated during early learning[48], particularly when motor imaginary is used[74]. In fact, athletes (i.e., experts or high performers) have been shown to activate the DN when employing strategies that decrease variability, resulting in stable performance during movement[68]. The phenomenon, known as in the zone[63], has been linked to the DN activation with consistent performance associated with preparedness[75,76] and vigilance[77]. Hence, activation of DN indicates those subjects are prepared for the chance of a perturbation. Taken together, these findings suggest that high performers react to uncertainty by heightening vigilance through activation and connectivity in DN in conjunction with the VAN and the visual network.

Observing movement variability in the form of motor error is common in motor control, with paradigms typically focused on aspects of motor learning. Numerous motor control studies have found—directly or indirectly—the involvement of regions in DAN and working memory[53,56,78]. In fact, our results align with classic motor control reports when considering their results in terms of networks. For example, Diedrichsen et al.[79] identified neural correlates of error in

DAN and visual networks, represented by SPL and POS respectively, identical to ours. Gnadt & Anderson[52] observed persistent activity in IPS (hub for DAN) in relation to target location during the delay in motor planning, connecting their results to memory. In a study directed towards large-scale brain networks, DAN activation during early learning was correlated with decreased error rate, which they related to active visuospatial attention when first learning the sequence[48].

This study highlights the complexity of behavioral and neural data as well as how challenging it is to disentangle internal states from other processes, nevertheless, we acknowledge several limitations inherent to our approach. First, it is possible that some of our neural data include results from epileptic brain regions in which activity could differ from comparable regions in healthy humans, despite precautions we took to minimize this possibility as discussed in Methods. The effects of anti-epileptic medications are another confounding variable that could have influenced the magnitude of the results, though subjects ceased their medications during clinical investigation. At this time, the only ethical method to record from the brain necessary for our study using SEEG depth electrodes in humans is while they are implanted for clinical purposes. Second, our behavioral data was limited by the design of the motor task and trial conditions, including two speeds, four directions, and few perturbation trials. Since internal states rely on the accumulation of history across trials, the validity of a state such as the perturbed state does not depend on the number of perturbed trials but rather the overall number of trials. For example, if no trials were perturbed, then the perturbed state would remain 0. Future experiments could explore other trial conditions, like introducing obstacles into the task space or other stimuli such as audio, or varying the probability of perturbations. One could also imagine designing a study that picks trial conditions to produce the desired variability from a subject based on model inferences about their internal states. Third, we focused on using a simple modeling approach, which raises the possibility that a key factor in a subject's behavioral variability may be absent from our model. Contrarily, this simplicity allows for other variables, such as other trial conditions or internal states, to be easily designed and integrated to create a model for a variety of behavioral tasks. Finally, we want to emphasize that the performance-related results should be taken as preliminary as the sample sizes for these statistics were limited to the number of subjects implanted in the same regions. Hence, studies with larger sample sizes are needed to make any conclusive statements.

In conclusion, our findings provide a fresh viewpoint for motor control research. Behaviors are readily measured every day from devices such as smartphones. Our results raise the possibility that the underlying history of measured behaviors could be used to make inferences about a person's brain state without needing to collect electrophysiological data, saving time and money in the health field for personalized medicine[80] or business ventures such as sports[81]. Future studies in motor control should consider the effect of these networks on motor control and should account for the effects of internal states as we found that they play a significant role in governing behavior and its variability.

## Methods

### Recording neural data from humans

Ten human subjects (seven females and three males; mean age of 34 years) were implanted with intracranial SEEG depth electrodes and performed our motor task at the Cleveland Clinic. These subjects elected to undergo a surgical procedure for clinical treatment of their epilepsy to identify an Epileptogenic Zone (EZ) for possible resection. Details of the handedness and clinical information of each subject are listed in Table 1. The study protocol, including experimental paradigms and collection of relevant clinical and demographic data, was approved by the Cleveland Clinic Institutional Review Board. Subject

criteria required volunteering individuals to be over the age of 18 with the ability to provide informed consent and able to perform the motor task. A data-sharing agreement between the Cleveland Clinic and Johns Hopkins University was approved by the legal teams of both institutions. Other than the experiment, no alterations were made to their clinical care. We excluded two additional subjects who attempted to perform the task but failed to complete it.

Each subject was implanted with 8–14 stereotactically-placed depth electrodes (PMT® Corporation, USA). Each electrode had between 8–16 electrode channels (henceforth referred to as channels) spaced 1.5 mm apart. Each channel was 2 mm long with a diameter of 0.8 mm. Depth electrodes were inserted using a robotic surgical implantation platform, (ROSA®, Medtech®, France) in either an oblique or orthogonal orientation. This procedure granted access to broad intracranial recordings in a three-dimensional arrangement, which included lateral, intermediate, and/or deep cortical as well as subcortical structures[82]. The day prior to surgery, volumetric preoperative Magnetic Resonance Imaging (MRI) scans (T1-weighted, contrasted with MultiHance®, 0.1 mmol kg$^{-1}$ of body weight) were obtained to plan safe electrodes trajectories that avoided vascular structures preoperatively. Postoperative Computed Tomography (CT) scans were obtained and coregistered with preoperative MRI scans to verify electrode placement postoperatively following implantation[82]. Electrophysiological data in the form of Local Field Potential (LFP) activity were collected onsite in the Epilepsy Monitoring Unit (EMU) at the Cleveland Clinic using the clinical electrophysiology acquiring system (Neurofax EEG-1200, Nihon Kohden, USA) with a sampling rate of 2 kHz referencing an exterior channel affixed to the skull. Each recording session was also determined to be free of any ictal activity.

## Inducing movement variability using our motor task

Our motor task was a center-out delay arm reach where subjects won virtual money by controlling a cursor on a screen to reach a target with an instructed speed despite a chance of encountering a random physical perturbation[72,83–85]. Subjects performed this task in the EMU using a behavioral control system, which consisted of three elements: a computer screen, an InMotion2 robotic manipulandum (Interactive Motion Technologies, USA), and a Windows-based laptop computer[83]. The computer screen (640 × 480 px) was used to display the visual stimuli to the subject. Subjects were seated ~60 cm in front of the screen. The robotic manipulandum allowed for precise tracking of the arm position in a horizontal two-dimensional plane relative to the subject. The subject used the robotic manipulandum to control the position of a cursor on the computer screen during the motor task restricted to a horizontal two-dimensional plane relative to themselves. The laptop computer ran the motor task using a MATLAB-based software tool called MonkeyLogic (version 2.72)[86].

During the session, subjects would complete as many trials as they could in 30 min. A complete trial consisted of eight epochs, each distinguishable with unique visual stimuli shown in Fig. 1a. Subjects began each trial with an instructed speed, indicating whether they were supposed to move fast or slow (Speed Instruction). Next, subjects moved their cursor to a target in the center of the screen (Fixation). Once centered, subjects were presented with a target in one of the four possible directions (Show Target). A random delay was applied here in which subjects could not move their cursor out of the center until cued to do so. This cue was signaled as the target changing color from gray to green (Go Cue). After their cursor left the center (Movement Onset), there was a chance that a constant perturbation would interrupt their movement. Subjects were still expected to reach the target with the correct speed despite the perturbation. Once they reached (Hit Target) and held their cursor in the target, subjects were immediately presented with feedback on their trial speed compared to the instructed speed (Speed Feedback). The reward they were shown depended on whether they matched the instructed speed or not (Show Outcome).

An image of an American $5 bill was presented for correct trials while the same image overlaid by a red X was presented for incorrect trials. It should be noted that subjects did not receive any monetary reward for participating in this task. Epochs are structured into traditional phases of motor control based on the design of the experiment. Planning includes Speed Instruction, Fixation, Show Target and Go Cue, Execution includes Movement Onset and Hit Target, and Feedback includes Speed Feedback and Show Outcome.

Subjects could fail a trial for any of the following reasons: not acquiring the center during Fixation, leaving the center before Go Cue, failing to leave the center after Go Cue, or inability to reach the target during Movement Onset. Regardless of the reason, the rest of the trial was aborted and subjects were presented with a red X before moving to the next trial.

We only used completed trials (i.e., trials in which the Speed Feedback was reached) for our study. Subjects were aware that perturbations would be applied. Additionally, subjects were allowed as much time as they wanted to practice the motor task before the session began, which included the speeds, directions, and perturbations.

At the end of each session, the session performance of each subject was calculated as 100*(Number of completed trials with correct speed)/(Number of completed trials). Session performance of 0% means the subject achieved the correct speed on none of their trials and session performance of 100% means the subject achieved the correct speed on all of their trials. To differentiate the performance of a trial (i.e., correct or incorrect) from the session (i.e., percent of correct trials), we refer to the former as trial performance.

There are three trial conditions that varied from trial-to-trial: the instructed speed, the instructed target direction, and the type of perturbation. Speed refers to the binary condition categorizing the instructed speed (fast, slow). Either speed was equally likely for each trial. In actuality, the categorical representation of speed translates to a range of values based on the percent of a subject-specific maximum speed measured during calibration; when they were told to move the cursor as quickly as possible from the center to a right target over five trials just before starting the experiment. Fast trials accepted 66.67 ± 13.33% and slow trials accepted 33.33 ± 13.33% of their calibration speed. Direction refers to the four possible locations of the target relative to the center of the computer screen (down, right, up, left). The probability of each location was equally like for each trial. Perturbation refers to the type of perturbation, if any, that was experienced during the trial (unperturbed, towards, away). Each trial had a 20% probability of a perturbation being applied with a random force between 2.5 to 15 N at a random angle, both selected from a uniform distribution. The perturbation was physically applied to the subject using the robotic manipulandum and would persist until the subject was shown their feedback. Perturbations can be categorized based on the angle it was applied relative to the target direction: towards or away. All other trials are unperturbed. The summary of the trial conditions experienced by each subject are listed in Supplementary Table 3.

In addition to trial conditions, we also tracked two continuous values that incurred movement variability from trial-to-trial: Reaction Time (RT) and Speed Error (SE). The RT (in seconds) was quantified as the time it took for a subject to move their cursor out of the center after Go Cue. The SE was quantified as the difference between the middle of the range of the instructed speed (0.33 or 0.67) and their trial speed. The trial speed was found by dividing the constant distance between the center and the target (in pixels) with the total time between Go Cue and Hold Target (in seconds), then scaling it by their calibration speed. The SE can take on a value between −0.67 to 0.67, where a positive SE means the subject moved slower than instructed (i.e., too slow), a negative SE means the subject moved faster than instructed (i.e., too fast), and a SE between −0.13 and 0.13 means the subject was within the acceptable range for the trial to be correct. The

summary of the RT and SE for each subject are listed in Supplementary Tables 1 and 2.

To test the main effects and interactions that subjects and trial conditions had on RT and SE, we constructed a multi-way ANOVA for each. For RT, we used a three-way ANOVA using subject ($n = 10$), speed ($n = 2$), and direction ($n = 4$) as the independent factors. For SE, we used a three-way ANOVA using subject ($n = 10$), the combination of speed and perturbation called type of perturbation ($n = 6$), and RT as the independent factors. Subjects were treated as a random variable for both tests and the RT factor in SE was treated as a continuous variable. Any initial results that indicated significant differences were followed up by a post-hoc Tukey's comparisons. The results of these tests are reported in Results and shown in Supplementary Tables 4–10.

### Estimating internal states to capture movement variability

We sought to construct a behavioral model to capture movement variability based on data collected during our goal-directed center-out delay arm reach motor task. The behavioral data consisted of any quantifiable measurements from the motor task, namely the trial conditions, RTs, and, SEs.

Our system follows the framework outlined in Fig. 2a. It takes on the structure of a state-space representation and consists of three basic elements for each trial $t$: outputs, inputs, and internal states. Based on the design of our motor task, we assume that a movement on every trial goes through two phases: planning and execution. This is represented by two boxes seen in Fig. 2a. The inputs of planning are speed and direction while the output is RT. The input of execution is perturbation as well as the RT from planning while the output is SE. Internal states are drawn as a black dashed line for illustrative purposes. They provide feedback for both planning and execution. This is because the internal states update is based on trial history (such as past performance or trial conditions). This information then flows through our system to affect the outputs. Though it is not labeled, the dotted line from planning to execution also carries the inputs from the planning system (speed and direction) to be available for the downstream system. Therefore speed and direction are also available for modeling SE. However, perturbation is not available for RT because perturbations happen after RT. However, the history of the trial conditions from previous trials is available through the internal states.

The outputs, RT and SE, are denoted $y_t^{RT} \in \mathbb{R}$ and $y_t^{SE} \in \mathbb{R}$ respectively. They are directly measured from the behavioral data during the motor task. The RT was normalized using the $z$-score before any modeling was performed so each subject followed a standard normal distribution (i.e., $N(0,1)$). The SE innately followed a continuous uniform distribution (i.e., $U_{[-0.67, 0.67]}$) and was not normalized. Refer to Supplementary Tables 1 and 2 for the summary of outputs. The inputs are the trial conditions of the motor task: speed, direction, and perturbation. They are also directly measured. They are described as categorical variables, denoting speed as $u_t^S \in \{fast, slow\}$, direction as $u_t^D \in \{down, right, up, left\}$, and perturbation as $u_t^P \in \{unperturbed, towards, away\}$. The internal states we defined are the error state and perturbed state, denoted $x_t^{SE} \in \mathbb{R}$ and $x_t^P \in \mathbb{R}$ respectively.

Our system is broken down into the phases of planning and execution (Fig. 2a). The behavioral outputs of planning and execution are RT[87] and SE[25], respectively. They are separated by a delay for maximal separation[88]. It is important to note that the states remain constant between planning to execution since neither state has information to update until after a movement is complete. Each phase is associated with its own mathematical function relating the outputs as a linear combination of states and inputs available on trial $t$. The task

begins with the planning phase, written as:

$$y_t^{RT} = \underbrace{\beta_0^{RT}}_{\text{Constant}} + \underbrace{\beta_{RT}^{SE} x_t^{SE}}_{\text{Error state}} + \underbrace{\beta_{RT}^{P} x_t^{P}}_{\text{Perturbed state}} + \underbrace{\sum_{s \in \{fast, slow\}} \beta^s \mathbb{1}(u_t^S = s)}_{\text{Speed}} + \underbrace{\sum_{d \in \{down, right, up, left\}} \beta^d \mathbb{1}(u_t^D = d)}_{\text{Direction}} + \underbrace{\epsilon_t^{RT}}_{\text{Noise}},$$

$$(1)$$

where $\epsilon_t^{RT}$ is an independent normal random variable with zero mean and variance $\sigma_{RT}^2 \in \mathbb{R} \geq 0$. In other words, it defines the output of planning on trial $t$ as RT and is the linear combination of a constant, error state, perturbed state, speed, and direction on trial $t$, scaled by their respective weights ($\beta$'s). This is followed by the execution phase written as:

$$y_t^{SE} = \underbrace{\beta_0^{SE}}_{\text{Constant}} + \underbrace{\beta_{SE}^{SE} x_t^{SE}}_{\text{Error state}} + \underbrace{\beta_{SE}^{P} x_t^{P}}_{\text{Perturbed state}} + \underbrace{\beta_{SE}^{RT} y_t^{RT}}_{\text{RT}} + \underbrace{\sum_{s \in \{fast, slow\}} \sum_{p \in \{unpert., towards, away\}} \beta^{s,p} \mathbb{1}(u_t^S = s)\mathbb{1}(u_t^P = p)}_{\text{Type of perturbation}} + \underbrace{\epsilon_t^{SE}}_{\text{Noise}},$$

$$(2)$$

where $\epsilon_t^{SE}$ is an independent normal random variable with zero mean and variance $\sigma_{SE}^2 \in \mathbb{R} \geq 0$. It defines the output of execution on trial $t$ as SE and is the linear combination of a constant, error state, perturbed state, RT, and the combination of speed and perturbation on trial $t$, scaled by their respective weights ($\beta$'s). By our definition, RT will always be available as an input for SE. Though speed is not a direct input to execution (Fig. 2a), it also carries over from planning. We found the combination of speed and perturbation captured SE well as compared to any other linear combination of trial conditions. This combination is supported by intuition as well as in literature[89]. On one hand, a slow trial with an away perturbation could help subjects reduce the magnitude of their SE by forcing them to move slower. On the other hand, a fast trial with an away perturbation could make it harder to match the speed, making a negative SE (too slow) more believable.

To capture the history of their performance of speed error during the task, we used SE to update the error state:

$$x_t^{SE} = \alpha^{SE} \underbrace{x_{t-1}^{SE}}_{\text{Error memory}} + \underbrace{y_{t-1}^{SE}}_{\text{Speed error}}.$$

$$(3)$$

The degree to which the previous state weighs into the current state is scaled by $\alpha^{SE}$, which ranges between 0 and 1. An $\alpha^{SE}$ closer to 0 means the error state will quickly decay to 0 on subsequent trials while a $\alpha^{SE}$ closer to 1 means the error state will retain its value, such as the case for subjects who carry information over from trial-to-trial. The input is the SE from the previous trial. It ranges in value between −0.67 and 0.67, where a positive SE ($y^{SE} > 0$) means they moved slower than instructed and a negative SE ($y^{SE} < 0$) means they moved faster than instructed. Therefore, a positive error state indicates the accumulation of trials that were slower than instructed whereas a negative error state indicates the accumulation of trials that were faster than instructed. The sign of the weights $\beta^{SE}$ in Eqs. (1) and (2) depicts how the behavior of a subject would respond to the error state. Take the case when the error state is positive (i.e., moving slower than instructed). A positive $\beta_{RT}^{SE}$ would increase the RT, thus subjects would react slower after trials in which they moved slower than instructed. Conversely, a negative $\beta_{RT}^{SE}$ would decrease their RT and subjects would react faster after trials in which they moved slower than instructed. For SE, a positive $\beta_{SE}^{SE}$ would increase the SE, meaning subjects would continue to move slower than instructed on subsequent trials. A negative $\beta_{SE}^{SE}$ would decrease the SE,

meaning subjects would move faster than instructed on subsequent trials.

To capture the effect of perturbations on their behavior, we used an indicator on whether the perturbation input detected a perturbation either towards or away to update the perturbed state:

$$x_t^P = \alpha^P \underbrace{x_{t-1}^P}_{\text{Perturb memory}} + \underbrace{\sum_{p \in \{\text{towards,away}\}} \mathbb{1}\left(u_{t-1}^P = p\right)}_{\text{Perturbation}} . \qquad (4)$$

The perturbed state receives a positive pulse when perturbation was applied, regardless of the type of perturbation that was applied. Therefore, the state only deviated from 0 when a perturbation was introduced. In the absence of a perturbation, the system would decay with a rate constant of $\alpha^P$, which ranges between 0 and 1 whereas a $\alpha^P$ closer to 0 indicates that the state will decay back to 0 by the next trial. A $\alpha^P$ closer to 1 means that subjects would carry over perturbations into subsequent trials through the perturbed state if it has not fully decayed to 0. Because of its structure, the perturbed state can only be positive or 0, where 0 means that perturbations do not affect behavior. The perturbed state affects behavior based on the sign of $\beta^P$ in Eqs. (1) and (2), so long as the perturbed state is not 0. In terms of RT, a positive $\beta_{RT}^P$ will increase the RT after a perturbation. Therefore, subjects with a positive weight will react slower than average after perturbation trials. A perturbation will also affect the SE based on the sign of $\beta_{SE}^P$. A positive $\beta_{SE}^P$ will increase SE after perturbation trials, i.e., subjects will move slower than instructed after perturbations. Subjects where both weights are positive suggest that they hesitated in response to recent perturbations captured by their perturbed state.

Unlike the other elements in our system, the internal states cannot be directly measured because they are subjective. They are an internal representation of the environment that an individual defines which evolves given new information. Instead, internal states are dynamically updated on a trial-by-trial basis by weighing their past states. Both must be estimated using a first-order state evolution equation, whose function is controlled by what is added to it in addition to their past states. The general solution is:

$$\hat{x}_t = \underbrace{\alpha^{(t-1)} \hat{x}_1}_{\text{Initial state}} + \underbrace{\sum_{i=1}^{t-1} \alpha^{(t-i-1)} u_i}_{\text{Cumulative state}} . \qquad (5)$$

Therefore, the states are not simply a weight of the previous trial but capture the accumulation of history from previous trials.

Thus, Equations (3)–(2) make up our system. But the system is not complete until fitting the models. Model fitting consisted of finding the combination of weights ($\alpha$'s and $\beta$'s) that minimized the root-mean-squared error between the observed and estimated outputs for each subject using all complete trials. First, the $\alpha$'s were found using a grid search between 0.01 and 0.99 at an interval of 0.01 with the initial conditions $x_1^{SE} = x_1^P = 0$ to estimate the internal states. Additionally, the internal states were normalized using the $z$-score so their weights could be compared across subjects. Then, the weights were found using methods of generalized linear model, which solves for the maximum likelihood estimation. The resulting states and weights were applied to estimate the outputs. The combination of weights with the largest two-tailed Pearson correlation value between observed and estimated outputs was selected as the final model. This process was repeated for each subject to create their custom model fitting, complete with their own individual evolution of internal states. Supplementary Table 11 contains the weights of the final model for each subject.

To show the adequacy of our model, we also built a subject-specific linear model that relied only on trial conditions. The linear model was a simple linear regression between the outputs (i.e., RT and SE) and inputs (i.e., speed, direction and RT, speed, perturbation). All models were evaluated using the coefficient of determination, deviance, and 10-fold cross-validation for comparison. The coefficient of determination was used to quantify how much the variability of the outputs can be explained by the inputs. It ranges in value between 0 and 1, where 1 means that the estimated outputs completely accounts for all the variability of the observed outputs (Fig. 2e). The deviance was used to compare the error of each model. This can be helpful as an absolute measurement of goodness-of-fit to compare models by measuring the trade-off between model complexity and goodness-of-fit. Deviance can be any positive value, where a deviance of 0 means that the model describes the observed outputs perfectly (Supplementary Fig. 6c, d). Cross-validating our internal state models proved difficult, as the internal states are history-dependent. Since the focus of our paper was on how internal states influence behavior as opposed to trying to predict behavior, our final models were fitted using all data. We implemented a 10-fold cross-validation using the internal states fitted using all data (Supplementary Fig. 6e, f) to show that adding internal states improved model performance.

## Neural data preprocessing

We used spectral analysis to preprocess the LFP activity from each channel. First, a notch filter was applied to the raw voltage data using notches located at the fundamental frequency of 60 Hz and its higher harmonics with the bandwidth at the -1 dB point set to 3 Hz. Next, the oscillatory power was calculated using a continuous wavelet transform with a logarithmic scale vector ranging 1–200 Hz and complex Morlet wavelet with a default radian frequency of $\omega_0 = 6$. The resulting instantaneous power spectral density was divided into overlapping time bins using a window of 100 ms every 50 ms. All overlapping time bins were averaged together and labeled with the last time index corresponding to that window. Finally, the averaged power spectral density was normalized to equally weigh all frequency bins by taking the $z$-score of the natural logarithm of the power in each frequency bin over the entire recording session time. All channel recordings were visually inspected for artifacts before and after preprocessing. Examples of artifacts include broadband effects, abnormal bursts of power, and faulty recordings. Any channels with artifacts were disregarded for the entire session. Figure 4d shows the result of the spectral analysis for a channel as a spectrogram, where the color of each pixel represents the normalized power between -3 and 3 indexed by the color bar at a specified frequency and time.

The results of our analysis depend heavily on how channels are aggregated using their anatomical labels. Therefore, it was important to ensure that their labels were unbiased across subjects. We applied a semi-automated electrode localization protocol to determine the coordinates of each channels per subject by fusing their preoperative MRI with postoperative CT[90,91]. This protocol also labeled each channel using an anatomical atlas[92] and a large-scale functional brain network atlas[93] based on their coordinates onto a subject-specific cortical parcellation[94]. The anatomical labels were validated by a clinician. To visualize the coverage of channels across the populations, their coordinates were warped from the subject's native space to the standard Montreal Neurological Institute (MNI) atlas space[95]. Figure 4a shows the all channels of all subjects projected onto the template brain called `cvs_avg35_inMNI152` from Freesurfer[94].

## Identifying brain regions that encode internal states

After neural data preprocessing, we used an unsupervised paradigm to identify where internal states are encoded in the brain of the population. We accomplished this using a non-parametric cluster statistic[36]. A description of how this method was applied to similar data has been detailed by our group previously[20]. Here, we used a two-tailed permutation test with $N = 1000$ and a significance threshold of $\alpha = 0.05$. In

short, the procedure statistically identifies windows of time and frequency where the LFP (measured as power in the spectrogram) of a region significantly correlates with an internal state across trials in the population (Fig. 4d). These regions were formed by aggregating the anatomical labels of the channels from all subjects. The result is windows of time and frequency, known as clusters, across brain regions. Each cluster also has a cluster statistic, where higher magnitudes are favored as having a stronger cluster[36].

Clusters that were too small (i.e., had windows <250 ms in time, one octave in frequency, or window area was smaller than the minimum time and frequency windows specified) were discarded. We also discarded regions that had less than two subjects contributing to the cluster. A false discovery rate of $q = 0.015$ was then applied to the cluster statistic to correct for multiple comparisons between regions and epochs. Clusters are confined to predefined windows of time set by the epoch it was recorded from for the analysis. However, since neural activity is continuous, information it may be encoding could carry over from one epoch to the next. Therefore, clusters in the same region with overlapping frequency bins across epochs were grouped for further analysis. This also meant that a region could come up multiple times, such as the case if separate clusters were found in different frequency bins in the same epoch.

Each group of clusters spans across multiple frequency bins. For example, Fig. 4d shows the outline of a cluster that includes frequency bins between 60–200 Hz over the statistical map. Therefore, each group of clusters was mapped to frequency bands as commonly defined in literature based on their frequency bins: delta (1–4 Hz)[96], theta (4–8 Hz)[96], alpha (8–15 Hz)[96], beta (15–30 Hz)[96], low gamma (30–60 Hz)[97], high gamma (60–100 Hz)[97], and hyper gamma (100–200 Hz)[98]. If the group of clusters spanned multiple frequency bands, then the band that made up the majority of the area of the group of clusters was prioritized.

Further, we observed two distinct temporal patterns of activity that described each group of clusters; persistent or phasic. Persistent activity refers to a group of clusters whose activity stretched across all epochs during a trial. Phasic activity refers to a group of clusters whose activity only appeared during specific epochs of a trial related to the movement phase (i.e., planning, execution, feedback).

## Calculating encoding strength

We hypothesized that neural activity in encoding networks of subjects will modulate with internal states based on session performance. To test this, we quantified how well a region covaries with the internal state using encoding strength and compared it to session performance across subjects. We calculated the encoding strength of group of clusters for each subject by replicating the procedure from the nonparametric cluster statistic. For each group of clusters, we first averaged the neural activity in its time-frequency window across the epochs it spans for each channel, trial, and subject (Fig. 4b, c). Next, we found the magnitude of the two-tailed Spearman's correlation value between this averaged neural activity and the internal state across all trials for each channel and subject (Fig. 4e). Finally, we averaged these correlation magnitudes across channels in a subject with the same group of clusters. This value is the encoding strength of the group of clusters for a subject of a region. Because it is derived from the magnitude of the correlation, it takes on a value between 0 and 1, where 1 means the averaged neural activity in the cluster exactly follows the internal state across trials.

To identify regions that encode internal states and performance, we correlated the encoding strength of each group of clusters to the session performance across subjects. This allowed us to identify the regions that not only encoded the internal states but also related to session performance (i.e., subject variability). These performance-related regions were those whose two-tailed Pearson correlation value exceeded 0.75 or p-value was significant ($p < 0.05$). We did not rely

solely on significant p-values because the largest possible sample size (i.e., $n \leq 10$) was small. Further, we used the magnitude of correlation value because we were interested in the relationship between encoding strength and session performance, not the direction of encoding (i.e., positive or negative), aggregated in Fig. 5b, e. Because we were focused on session performance, we rejected regions that did not have at least two subjects whose performance was either above and below average performance (51%).

A table of these regions, and the large-scale brain network they belong to, can be found in Tables 2 and 3 for error and perturbed state, respectively. They are also displayed on an inflated brain template in Fig. 5a, d, where gyri and sulci can be visualized together.

## Calculating connectivity strength

We hypothesized that differences in functional connectivity within and between networks that encode internal states account for learning strategies across the spectrum of varying session performance. To test this, we compared functional connectivity with session performance across subjects. Functional connectivity is defined as dynamic connections between neuronal populations through oscillatory activity[99]. There are many ways to calculate functional connectivity[99]. We chose to use cross-correlation using a lag of 0, which essentially is the two-tailed Pearson correlation value[37]. This value is what we call the connectivity strength. We calculated connectivity strength within each internal state at three levels between pairs: channel, subject, and population. Pairs refer to the fact that connectivity strength measures the interaction of two regions.

To calculate connectivity strength, we began by averaging the neural activity of each group of clusters using the results from the nonparametric cluster statistic for each channel, trial, and subject (Fig. 6a) to get the signal of average normalized power across trials. Then, we calculated the magnitude of the two-tailed Spearman's correlation value between all these signals within each subject (Fig. 6b). This is channel connectivity strength.

To calculate subject connectivity strength, the channel connectivity strengths of unique pairs were averaged within each subject. For example, to find the subject connectivity strength of IPS R and MTG R for subject 6, we would average the values from Fig. 6b to get a scalar value shown on the y-axis in Fig. 6c. Because it is derived from the magnitude of the correlation, the subject connectivity strength takes on a value between 0 and 1, where 1 means the activity of the pair is perfectly correlated across trials.

To identify preliminary pairs of regions whose connectivity encodes the internal states and performance, we related the subject connectivity strength to session performance for each pair across subjects the two-tailed Pearson correlation value (Fig. 6c). Any pair that did not include both a subject with above and below average session performance was excluded. We also excluded pairs that were not represented by at least three subjects. We refer to this as performance connectivity strength, which varied between -1 and 1 (Supplementary Fig. 9a, d). Supplementary Fig. 9b, e, c and f show the p-values and sample sizes used when calculating the performance connectivity strength. However, we were only interested in examining pairs of regions whose performance connectivity strength was strongly positive (i.e., higher performance with higher subject connectivity), which we defined as values that were >0.5. Pairs with values <0.5 were truncated in the main figure. These pairs are represented in Fig. 6d, e for error state and perturbed state, respectively, using directed graphs.

For population analysis, we first transformed the subject connectivity strength using Fischer's z transformation[100] before averaging pairs across subjects. After averaging, the values were transformed back into correlation magnitudes using Fischer's z transformation[100] due to our small sample size[101]. These values represent the population connectivity strength. Any pairs that had fewer than two subjects were excluded. Likewise, connectivity

strength on the diagonal (i.e., autocorrelations) was ignored for both subject and population connectivity strengths before averaging. See Supplementary Fig. 10 for a summary of the population connectivity strength across all encoding regions, including the sample sizes.

### Reporting summary

Further information on research design is available in the Nature Portfolio Reporting Summary linked to this article.

## Data availability

The raw SEEG data are protected and are not available due to restrictions on data sharing from Cleveland Clinic. The processed data that support the findings of this study are available on Johns Hopkins Research Data Repository with the identifier doi:10.7281/T1/PIVKJ7[102]. The data generated in this study are also provided in the Source Data file. Source data are provided with this paper.

## Code availability

The code used to generate figures for this paper is available on Johns Hopkins Research Data Repository with the identifier https://doi.org/10.7281/T1/PIVKJ7 by running the function MAIN.m[102]. The code for the nonparametric cluster statistic and semi-automated electrode localization is publicly available through FieldTrip (version 20191008) the identifier https://doi.org/10.1155/2011/156869[90]. The code for cortical surface extraction is publicly available through Freesurfer software suite (v.6.0.0) using the identifier https://doi.org/10.1016/j.neuroimage.2012.01.021[94].

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

## Acknowledgements

The authors would like to thank Juan Bulacio, Jaes Jones, Hyun-Joo Park, and Susan Thompson who collected, deidentified, transferred, and anatomically labeled the neural data as well as Matthew S.D. Kerr, Kevin Kahn, and Matthew A. Johnson who designed and collected the behavior data. We also thank Amy J. Bastian and Vikram S. Chib for our discussions during thesis meetings over the years that helped propel this work forward. Emery Brown additionally aided us during the review process by providing his statistical perspective on our analysis, which we thank him for. Finally, thank you to Jesse A. Smith for his extensive help proofreading and overarching moral support. Part of this research project was conducted using computational resources and scientific computing services at the Maryland Advanced Research Computing Center (MARCC). This work was supported by National Science Foundation grant (EFRI-MC3: #1137237) awarded to J.A.G.-M., J.T.G., and S.V.S.

## Author contributions

M.S.B., J.A.G.-M., J.T.G., and S.V.S. designed the research; M.S.B., J.A.G.-M., J.T.G., and S.V.S. performed the research; M.S.B, P.S., and S.V.S. contributed the analytic tools; M.S.B., P.S., and S.V.S. analyzed the data; M.S.B., P.S., Z.B.F., J.A.G.-M., and J.T.G. provided the resources and data curation; K.E.C, J.A.G.-M., and S.V.S. supervised the work; and M.S.B., P.S., Z.B.F., J.T.G, K.E.C., J.A.G.-M., and S.V.S. wrote the paper.

## Competing interests

The authors declare no competing interests.
