## [Peer Review File · Nature Communications]

Internal states as a source of subject-dependent movement variability are represented by large-scale brain networksREVIEWER COMMENTS

Reviewer #1 (Remarks to the Author):

The authors sought to link latent factors ("internal states") to motor performance via neural activity. 10 participants with stereo electroencephalographic (SEEG) recordings performed a reaching task using a robotic manipulandum. The speed (fast, slow) with which participants were to complete their reach varied across trials. Additionally, on 20% of trials a perturbation was applied either toward or away from the target. Through state-space representation modeling, the authors identified two key internal states: the error state, which tracks past errors and the perturbed state, which tracks past perturbations, to predict variability in both reaction times and speed errors -- how much the actual speed on a given trial deviated from the target speed. The authors claim that spectral signals in the dorsal attention network (DAN) track the error state whereas the default mode network (DN) tracks the perturbation state and conclude that these networks regulate motor strategy through internal states.

Although the question of how past experience influences motor learning would be of interest to the field, the interpretations made in this manuscript are not supported by the analyses and results. The limited number of participants - understandable due to the population - severely limits conclusions drawn regarding across subject variability. A number of methodological details are absent or challenging to understand. Below I outline my concerns.

1. Absence of direct statistical tests.

The authors make claims regarding the neural substrate of error and perturbation states without the appropriate statistical tests. "Remarkably, we then found that these internal states were linked to encoding in large-scale brain networks, DAN and DN, respectively." (Page 22) and "For the first time, our results implicate DAN as a network as encoding tracking history" (Page 23). However, to make such claims, a direct statistical test must be performed on DAN vs. at least one other region, and the other region cannot have been derived from visual inspection of a figure. The presence of an effect in a particular region does not indicate that the effect is specific or distinctive to that region. The same is true for frequency specificity; the authors highlight both high gamma (100-200 Hz) and low frequency (<15Hz) effects without proper statistical comparisons across frequencies.

2. An overwhelming lack of statistical reporting and claims that appear to be made on the basis of visual inspection rather than appropriate statistical testing.

Every statistical test reported in the manuscript needs to include report of a test statistic, degrees of freedom, p-value, and an effect size. These can be reported in a table but it is insufficient to report p-values alone. Additionally, post-hoc tests must be reported for any follow up analyses. Correction for multiple comparison must also be performed and reported. Many of the reported results cannot be evaluated due to insufficient statistical information. For example, the authors report, "Indeed, subjects reacted more quickly for fast trials than slow trials ($p = 0.014$, ANOVA). They also reacted more slowly for trials with an upward motion to the target ($p = 0.013$, ANOVA)." (Page 8). F-statistics, degrees of freedom, and effect sizes must be reported. What type of ANOVA was used and what were the factors? Furthermore, it seems that these results should have been found via a post-hoc t-test following a multi-way ANOVA. Means and standard deviation of the individual conditions (e.g. RT for upward trials) should also be reported. There is an over reliance on figures to support claims rather than statistical tests (e.g. "Visually, the estimate follows key features, including sudden jumps between trials and gradual changes such as between trials 100 and 125. For example, the estimate on trial 109 (black triangle) matches what was observed, which was that subject 6 reacted faster than average."). Multiple figures also specifically highlight Subject 6 and it is unclear how/why this participant was selected.

3. The connectivity analysis is confusing and challenging to interpret.

The authors describe the connectivity analysis as

"To calculate connectivity, we began by averaged the neural activity of each group of clusters using its time-frequency window across the epochs it spans for each channel, trial, and subject. Then, we found the magnitude of the Pearson correlation value between each pair the averaged neural activity of group of clusters and channels for each subject across trials. Pairs of channels in the same group of clusters were then averaged within each subject. These values represent the connectivity strength of the pair of regions for each subject."

It's not clear what is being correlated exactly. The authors use the term "pair of regions," but it is unclear what that means or how regions are defined. How many pairs of regions were identified for each participant and how many correlations were performed? I am concerned about the number of statistical tests that were performed without correction for multiple comparisons. Furthermore, the connectivity analysis was only performed on top performers. This choice was insufficiently justified (Page 19, lines 6-8), limits the ability to generalize the current findings beyond this group of participants, and renders claims such as "...exhibits stronger functional connectivity for top performers" (Page 3, lines 2-3) inaccurate since there are no direct comparisons.

4. The confluence of multiple factors makes any results difficult to generalize.

Although intracranial EEG can present a unique opportunity and researchers leveraging such data are limited by clinical necessities, the choice to investigate large scale networks seems suboptimal in a small population with (necessarily) variable recording locations. The number of subjects with a recording in any given region is small (the maximum appears to be 6). The number of perturbation trials is low, and appears to be imbalanced when separately considering toward vs. away, making fitting of the perturbation state more challenging. Dividing the participants into top and bottom performers further reduces the sample size and is hard to justify when the original N is already so low. The authors state that "Top performers benefited the most from adding internal states to RT" but fail to mention that RT is much better modeled for top performers (~20-60% coefficient of determination) than for bottom performers (~3-5% coefficient of determination) in both models, including the model without states added. In general it seems that more participants would be needed for many of the claims that are made.

5. Inappropriate frequency band selection.

The authors state, "The frequency band was identified by matching the frequency bins to frequency bands commonly defined in literature" (Page 46). It's unclear what this means, exactly, but it is not appropriate to define bands by visual inspection of time-frequency spectrograms. Bands should be defined a priori without respect to the current data.

Reviewer #2 (Remarks to the Author):

Thank you for the opportunity to review "Internal states as a source of subject-dependent movement variability and their representation by large-scale networks" by Breault and colleagues. The manuscript describes an SEEG study looking at how movement variability is encoded in the brain. The authors first fit an internal state model to participant behavioural data in a reaching task and find evidence of two states, related to errors and perturbations. They then localize these states to the well-studied dorsal attention and default mode networks.

The study addresses an important question that is likely to be of wide interest to the field. The authors carefully triangulate towards a coherent explanation of how internal states govern movement variability, shedding light on the neural mechanisms underlying speed-accuracy trade-offs. However, as I outline below, there are several conceptual and methodological concerns that should be addressed before recommending publication.

1. "Such reflections form latent factors called internal states that induce variability of movement and behavior to improve performance." This is minor, but there is a general tendency in the narrative, particularly in the Abstract and Introduction, to talk about internal states as a real biological entity. I think this is misleading, because all work on states has the starting assumption that ongoing behaviour/neural activity can be partitioned into states. I don't think the narrative would suffer if the authors discussed internal states more as a theoretical and methodological construct, rather than as a real biological phenomenon.

2. There is a considerable literature on the role of signal variability in noninvasive human imaging that could potentially be relevant , e.g.

Garrett, D. D., Samanez-Larkin, G. R., MacDonald, S. W., Lindenberger, U., McIntosh, A. R., & Grady, C. L. (2013). Moment-to-moment brain signal variability: a next frontier in human brain mapping?. *Neuroscience & Biobehavioral Reviews*, 37(4), 610-624.

McIntosh, A. R., Kovacevic, N., & Itier, R. J. (2008). Increased brain signal variability accompanies lower behavioral variability in development. *PLoS computational biology*, 4(7), e1000106.

3. There is also a large literature on dynamic states in human imaging, e.g.

Lurie, D. J., Kessler, D., Bassett, D. S., Betzel, R. F., Breakspear, M., Kheilholz, S., ... & Calhoun, V. D. (2020). Questions and controversies in the study of time-varying functional connectivity in resting fMRI. *Network neuroscience*, 4(1), 30-69.

4. Why standard deviation as opposed to the coefficient of variation, which takes mean scaling differences into account?

5. My main methodological concern is the arbitrary (mean) splitting of participants into top and bottom performers. This choice is unnecessary and potentially suboptimal as it transforms an inherently continuous variable into a categorical one. All the analyses could then be performed using correlations (with bootstrapping) rather than ANOVAs.

6. Connectivity estimated as correlated power. Why not use an established method that focuses on coherence/synchrony or on correlating amplitude envelopes?

7. Is there any cross-validation for the fitted internal states model? The authors compare their model with a pure linear model and show that the former is better, but is there any way to demonstrate that the fitted model helps to predict future values of RT?

8. "First, a Notch filter was applied using notches located at the fundamental frequency of 60 Hz with bandwidth at the -1 dB point set to 3 Hz". The word "notch" does not need capitalizing. Also, did the authors include filters at subsequent harmonics of 60Hz?

9. The authors assign regions to frequency bands, but does this method account for the fact that the power spectrum has a $1/f$ shape and low frequencies will naturally tend to have greater power than higher frequencies?

10. The other major methodological concern is how network specificity is established for the reported effects (e.g. localizing states to DAN or DMN), Namely, the comparison was clearly made visually/qualitatively, but the different networks all have different size, spatial coverage, etc. This means that some networks are more likely to get "hits" than others. The authors should confirm that each network is enriched for a particular state by computing the mean in the network and then comparing this to a null distribution where network labels are permuted.

Reviewer #3 (Remarks to the Author):

The main topic addressed by this paper is to examine how the brain represents internal states during motor processing, and to reveal the neural substrate of these patterns. Specifically, this paper utilizes state-space models to link internal states (past errors and perturbations) with motor performance and neural activity, and to identify variations in performance across subjects and trials. While performing a goal-directed center-out reaching task, ten human subjects' electrophysiological data and behavioral data were collected. The paper concluded that large-scale brain networks in the dorsal attention and default networks reflect the neural basis of strategy to regulate movement variability through internal states (past error and perturbation states, respectively) to improve motor performance.

Overall, the idea of examining the neural basis of internal states in regulating an individual's motor performance is interesting. The paper is well written and has many innovative ideas. However, my enthusiasm for the paper is lowered the impact of the neural data analyses is rather limited. The neural power analyses, which found signals at particular frequencies that correlated with aspects of behavior, were not described in a way that lets the reader make specific inferences about the functional contributions of those regions to behavior. The synchrony analyses were improved in this way, but still,

there was not enough detail to make a compelling case about the functional or mechanistic roles of particular regions or networks to support motor processing. Further, I have significant issues with the methods and data analysis that the authors presented.

Specific critiques:

The behavioral data were not presented with enough detail for me to understand the differences across good versus bad subjects, as well as to explain differences in the subjects that the model fit well versus poorly. I would suggest expanding figure 1 and the related sections of the text to separately analyze and interpret the behavioral characteristics of the group data for the subjects with different levels of absolute performance and for those who the model did or did not fit well.

The neural data analyses for figures 4, 5, and 6 are not illustrated with adequate statistical justification. I could not tell which neural signals correlated with behavior in a statistically robust fashion, and there was a lack of adjustment for multiple comparisons. Overall this section of the paper was largely qualitative, which is inadequate. This section of the paper should rigorously show which neural signals, across frequencies, correlated with behavior, and include both within- and across subject comparisons.

The analysis described in Figure 2 was hard to understand because it was focused so much on just subject 6.

The behavioral data in Figure 2 are hard to interpret because panels B-D are visually very dense. Panel E seems to indicate that the model fit quality is bimodal, with a cluster of four subjects who have four model fits and others who show rather good model fits. Does this undermine the results?

Figure 4C is hard to interpret because the statistic is calculated based on the correlation but the colors reflect a different value, mean power. It would be much easier to interpret this plot if the same value was plotted as indicated in the statistic.

Figure 6 is hard to interpret, is there a more intuitive or data-driven way to order the labels around the plots?

The analysis on Page 20, Line 30 and Fig 6 could benefit from a more quantitative comparison between network connectivity during “error state” and “perturbed state” rather than qualitative comparisons. There are various quantitative measures that the authors might use for this network comparison

Perhaps it is a minor point, but I found the conclusion of the paper confusing. How are the results relevant for the type of behavioral data that could be collected from smartphones? There might be an interesting idea here but it needs more length to spell out.

Please explain in the text that discuss in the text why was it was valid for the subjects to be divided based on a median split of RT? Some “top performers” may be doing relatively poorly in the task.

Reviewer #1 (Remarks to the Author):

The authors sought to link latent factors ("internal states") to motor performance via neural activity. 10 participants with stereo electroencephalographic (SEEG) recordings performed a reaching task using a robotic manipulandum. The speed (fast, slow) with which participants were to complete their reach varied across trials. Additionally, on 20% of trials a perturbation was applied either toward or away from the target. Through state-space representation modeling, the authors identified two key internal states: the error state, which tracks past errors and the perturbed state, which tracks past perturbations, to predict variability in both reaction times and speed errors -- how much the actual speed on a given trial deviated from the target speed. The authors claim that spectral signals in the dorsal attention network (DAN) track the error state whereas the default mode network (DN) tracks the perturbation state and conclude that these networks regulate motor strategy through internal states.

Although the question of how past experience influences motor learning would be of interest to the field, the interpretations made in this manuscript are not supported by the analyses and results. The limited number of participants - understandable due to the population - severely limits conclusions drawn regarding across subject variability. A number of methodological details are absent or challenging to understand. Below I outline my concerns.

Authors: We thank the reviewer for their helpful comments and positive feedback. As detailed below we have revised the manuscript to clarify the rigorous statistical methods that were applied in our analyses and made changes in the main document as well as supplemental material. In addition, we provide a point-by-point response to each of the reviewer's specific comments and indicate where we revised the manuscript.

1. Absence of direct statistical tests.

The authors make claims regarding the neural substrate of error and perturbation states without the appropriate statistical tests. "Remarkably, we then found that these internal states were linked to encoding in large-scale brain networks, DAN and DN, respectively." (Page 22) and "For the first time, our results implicate DAN as a network as encoding tracking history" (Page 23). However, to make such claims, a direct statistical test must be performed on DAN vs. at least one other region, and the other region cannot have been derived from visual inspection of a figure. The presence of an effect in a particular region does not indicate that the effect is specific or distinctive to that region. The same is true for frequency specificity; the authors highlight both high gamma (100-200 Hz) and low frequency (<15Hz) effects without proper statistical comparisons across frequencies.

Authors: Thank you for pointing out that we were not clear about the rigor of our neural analysis. To clarify, all of our neural results were derived using an unsupervised and data-driven approach, which were supported by statistical tests at all steps. First, the raw SEEG data was preprocessed into spectral data of normalized power in the time and frequency domains. The spectral data were then sliced into trials, which were then sliced into epochs. Next, we used a non-parametric cluster statistical test (Maris & Oostenveld, 2007) to find windows of time and frequency in which the spectral power correlated with each internal state across trials for each region across the population. This method is agnostic to time and frequency and avoids the burden of needing to correct for multiple comparisons at this level. These windows are called clusters, and each has an associated cluster statistic (Figure 4c). Finally, we corrected for multiple comparisons (one comparison for each brain region for each epoch for each internal state) using a false-discovery rate (FDR) of $q=0.015$. We hope this summary helps clarify our procedure and have revised the manuscript as detailed below.

Specifically, we now clarify that the results that we report in our paper are only those that pass the FDR above (see Tables 2 and 3 for statistics from nonparametric cluster analysis). As a result, this approach narrowed down our scope from 40 possible regions to just 12 for the error state and 14 for the perturbation state. Based on this data-driven approach, we noticed that the error state was populated by regions in DAN while the perturbation state was populated by regions in DN (Figure 5a-b).

Thus, our approach was entirely reliant on our statistical approach, however, we appreciate that this was not explained. To be more clear, we have revised our manuscript as follows:

1. Page 16–17:

- “To identify the neural correlates amongst these regions with an **unsupervised and data-driven method**, we used a non-parametric cluster statistic [36] (see Methods) between the spectral data of each region and each internal state across the population (Fig. 4b). This method finds windows of time (relative to epoch onset) and frequency (between 1 and 200 Hz) in which the spectral power of a region significantly correlates with the internal state across trials as demonstrated in Fig. 4c. **The statistic provided us with two sets of neural correlates, one for each internal state.**”
- “Our modeling results showed that subjects weighed internal states differently, primarily based on their session performance. We suspected that this would be reflected in the brain for each subject by how well these regions encoded the states through the strength of their neural correlates. **The degree to which a subject encodes an internal state in a region, which we called the encoding**

strength, was quantified by correlating the average power within the time-frequency window (from the population statistic) to the state on a trial-by-trial basis (see Methods). Fig. 4d shows an example of how the encoding strength is obtained for a channel in subject 6 using the neural correlate from Fig. 4c.

2. Figure 4, panel b,c,d

2. An overwhelming lack of statistical reporting and claims that appear to be made on the basis of visual inspection rather than appropriate statistical testing.

Every statistical test reported in the manuscript needs to include a report of a test statistic, degrees of freedom, p-value, and an effect size. These can be reported in a table but it is insufficient to report p-values alone. Additionally, post-hoc tests must be reported for any follow up analyses. Correction for multiple comparisons must also be performed and reported. Many of the reported results cannot be evaluated due to insufficient statistical information. For example, the authors report, "Indeed, subjects reacted more quickly for fast trials than slow trials ($p = 0.014$, ANOVA). They also reacted more slowly for trials with an upward motion to the target ($p = 0.013$, ANOVA)." (Page 8). F-statistics, degrees of freedom, and effect sizes must be reported. What type of ANOVA was used and what were the factors? Furthermore, it seems that these results should have been found via a post-hoc t-test following a multi-way ANOVA. Means and standard deviation of the individual conditions (e.g. RT for upward trials) should also be reported. There is an over reliance on figures to support claims rather than statistical tests (e.g. "Visually, the estimate follows key features, including sudden jumps between trials and gradual changes such as between trials 100 and 125. For example, the estimate on trial 109 (black triangle) matches what was observed, which was that subject 6 reacted faster than average."). Multiple figures also specifically highlight Subject 6 and it is unclear how/why this participant was selected.

Authors: Thank you for your suggestions. We agree that our manuscript required more detailed reporting of statistics. We have directly addressed these concerns in our revised manuscript. Specifically, we have added more detailed statistics in our text to include the statistical test, test statistic, degrees of freedom, p-value and effect size when appropriate (pages 7, 8, 9, 11, 12, 14, 18, 20) as well as added tables of statistics (ANOVA, post-hoc test, etc.) for behavior across conditions (Supplementary Tables 3, 4, 5, 6, 7). For example, the claim on page 8 stated as **"they reacted more slowly for trials when the target was up compared to when the target was down (post-hoc Tukey's: $p = 0.0037$) or right (post-hoc Tukey's: $p=0.016$)"** are now supported by post-hoc test reported in the text but detailed in Supplementary Tables 4 and 5.

The claim about sudden and gradual changes characterizing the conditions and states respectively can be captured using their averaged absolute second derivative (which we've added as Supplementary Figure 7a), in which conditions

had significantly larger values than states for both the RT (paired-sample t-test: $t(9)=4.96$, $p=0.0008$) and SE (paired-sample t-test: $t(9)=6.91$, $p=7.02e-05$). We added this result as well (page 11, **“Conditions accounted for the trial-by-trial changes and states accounted for the gradual changes across all subjects (Supplementary Figure 7a).”**)

Regarding subject 6, we now note that we focused on this subject to exemplify key points where we wanted the reader to gain intuition through visuals, which are backed up by the statistics as well. We added the following line to frame the reader for this thinking: pages 10—11 **“Though we used subject 6 to demonstrate the intuition behind our model, our general observations are applicable to the population, who are shown in Supplementary Fig. 1 – Supplementary Fig. 5.”** We have also incorporated other subjects into the interpretation of our model results in this section as well:

- Page 11, **“With the exception of RT for subject 1, we consistently found a significant relationship between the estimated and observed RT (Supplementary Fig. 1) and SE (Supplementary Fig. 2) for all subjects.”**
- Page 12, **“In RT, we also observed states that monotonically increased over trials for subjects 4, 5, 7, and 10 (Supplementary Fig. 3). This state reproduced subjects progressively reacting slower due to fatigue (Supplementary Fig. 1)”**
- Page 12, **“Based on the structure, we can also see what influenced the RT states to be monotonic for subjects 4, 5, 7, and 10. Supplementary Figure 5 shows that the primary contributor was the perturbed state.”**

3. The connectivity analysis is confusing and challenging to interpret.

The authors describe the connectivity analysis as

"To calculate connectivity, we began by averaged the neural activity of each group of clusters using its time-frequency window across the epochs it spans for each channel, trial, and subject. Then, we found the magnitude of the Pearson correlation value between each pair the averaged neural activity of group of clusters and channels for each subject across trials. Pairs of channels in the same group of clusters were then averaged within each subject. These values represent the connectivity strength of the pair of regions for each subject."

It's not clear what is being correlated exactly. The authors use the term "pair of regions," but it is unclear what that means or how regions are defined. How many pairs of regions were identified for each participant and how many correlations were performed? I am concerned about the number of statistical tests that were performed without correction for multiple comparisons. Furthermore, the connectivity analysis was only performed on top performers. This choice was insufficiently justified (Page 19, lines 6-8), limits the ability to generalize the current findings beyond this group of participants, and renders

claims such as "...exhibits stronger functional connectivity for top performers" (Page 3, lines 2-3) inaccurate since there are no direct comparisons.

Authors: We appreciate the reviewer's comment and agree that the original presentation of connectivity was unclear. We have since added panels b,c, and d to Figure 6 to help explain the method we used to arrive at the connectivity results.

Continuing from our response to concern 1, once the clusters (windows of time and frequency) were statistically identified, we calculated the average power in each cluster for each trial (visualized in Figure 4b, new Figure 6a). This gave us a signal of how the power in the cluster varied over trials for each channel in a region. This naturally led us to run a connectivity analysis between these signals, via Pearson's correlation, for each subject (new Figure 6b). After obtaining correlations between each pair of channels, the correlations for common pairs were averaged together to get subject connectivity (i.e., average all the correlations in Figure 6b). "Pairs" refers to the fact that connectivity measures the interaction of two channels. For example Figure 6a-c shows how we arrived at subject connectivity for the pair IPS R and MTG R for subject 6. We have revised the connectivity methods to make this clearer (page 51–53).

The number of correlations for each subject varied based on the number of regions that were identified using the non-parametric cluster statistic. But, to minimize the number of statistical tests, we only ran correlations within each internal state result. That is, we ran connectivity analysis for regions related to the error state and perturbed state separately. The following table summarizes the total number of correlations ran for each subject and internal state:

Subject	Error state	Perturb state
1	820	1830
2	45	528
3	15	91
4	595	630
5	0	0
6	153	325
7	66	153
8	3	28
9	91	300

10	703	2775
----	-----	------

which then averaged together to form unique pairs of regions for each subject into:

Subject	Error state	Perturb state
1	52	132
2	3	48
3	4	20
4	43	61
5	0	0
6	14	43
7	5	10
8	0	1
9	26	73
10	41	116

Though the connectivity in Figure 6d-e appears to only show top performers, the connectivity analysis on performance was actually performed across all subjects. Originally, we found that top performers had high connectivity between the pairs of regions shown in Figure 6d-e whereas the bottom performers had high connectivity between the pairs of regions in Supplementary Figure 9. Since then, we have removed the binary groups of subjects based on session performance and updated Supplementary Figure 9 to show all of the performance connectivity results.

4. The confluence of multiple factors makes any results difficult to generalize.

Although intracranial EEG can present a unique opportunity and researchers leveraging such data are limited by clinical necessities, the choice to investigate large scale networks seems suboptimal in a small population with (necessarily) variable recording locations. The number of subjects with a recording in any given region is small (the maximum appears to be 6). The number of perturbation trials is low, and appears to be imbalanced when separately considering toward vs. away, making fitting of the perturbation state more challenging. Dividing the participants into top and bottom

performers further reduces the sample size and is hard to justify when the original N is already so low. The authors state that "Top performers benefited the most from adding internal states to RT" but fail to mention that RT is much better modeled for top performers (~20-60% coefficient of determination) than for bottom performers (~3-5% coefficient of determination) in both models, including the model without states added. In general it seems that more participants would be needed for many of the claims that are made.

Authors: Thank you for your comment. Indeed, we are limited in the number of subjects, trials, regions, etc. Although the complete coverage cannot be obtained with any method of invasive monitoring, the spatial and temporal resolution with recordings from intracortical electrodes are optimal and highly superior to fMRI studies. These challenges are part of the limitations but also correspond to the main advantages of the method. Similar data has been highly accepted and published in literature [Sacre et al. 2019, PNAS]. We also addressed these concerns in the Discussion, under Study limitations:

- Page 31: **“At this time, the only ethical method to record from the brain necessary for our study using SEEG depth electrodes in humans is while they are implanted for clinical purposes.”**

As for perturbations, you are correct that the number of perturbation trials is low compared to unperturbed trials. If we were trying to predict whether a trial was going to be perturbed or not, then we would indeed be handicapped by the limited number of trials. However, the perturbed state is the accumulation of history fit using all trials. Therefore, the validity of the perturbed state does not depend on the number of perturbed trials. For example, if a patient never experienced a perturbation trial, then their perturbed state would be 0 for all trials. We also added the following to the discussion:

- Page 31 **“Second, our behavioral data was limited by the design of the motor task and trial conditions, including two speeds, four directions, and few perturbation trials. Since internal states rely on the accumulation of history across trials, the validity of a state such as the perturbed state does not depend on the number of perturbed trials but rather the overall number of trials. For example, if no trials were perturbed, then the perturbed state would remain 0.”**

Finally, we agree that dividing subjects based on performance was rather limiting and we no longer do that. As such we have redone Figures 1d, 2e, 3, 5c, 5e.

5. Inappropriate frequency band selection.

The authors state, "The frequency band was identified by matching the frequency bins to frequency bands commonly defined in literature" (Page 46). It's unclear what this means, exactly, but it is not appropriate to define bands by visual inspection of

time-frequency spectrograms. Bands should be defined a priori without respect to the current data.

Authors: We did not want to a priori any frequency bands because there is variability between subjects and their frequency bands [Haegens et al. 2014, NeuroImage]. This is why we chose a data-driven approach using the non-parametric cluster statistic, which was agnostic to frequency bands as it found windows of spectral power containing bins of time and frequency. Frequency bins were later mapped to frequency bands for interpretation only so that we could compare our results to existing literature that uses this language. A similar method of mapping frequency bins to bands was used for other work in our lab that used the nonparametric cluster statistic [Sacre et al. 2019, PNAS].

Reviewer #2 (Remarks to the Author):

Thank you for the opportunity to review “Internal states as a source of subject-dependent movement variability and their representation by large-scale networks” by Breault and colleagues. The manuscript describes an SEEG study looking at how movement variability is encoded in the brain. The authors first fit an internal state model to participant behavioural data in a reaching task and find evidence of two states, related to errors and perturbations. They then localize these states to the well-studied dorsal attention and default mode networks.

The study addresses an important question that is likely to be of wide interest to the field. The authors carefully triangulate towards a coherent explanation of how internal states govern movement variability, shedding light on the neural mechanisms underlying speed-accuracy trade-offs. However, as I outline below, there are several conceptual and methodological concerns that should be addressed before recommending publication.

Authors: We thank the reviewer for their positive feedback and helpful comments. In this revision, we provide a point-by-point response to each of the reviewer’s specific comments and indicate where we revised the manuscript.

1. “Such reflections form latent factors called internal states that induce variability of movement and behavior to improve performance.” This is minor, but there is a general tendency in the narrative, particularly in the Abstract and Introduction, to talk about internal states as a real biological entity. I think this is misleading, because all work on states has the starting assumption that ongoing behaviour/neural activity can be partitioned into states. I don’t think the narrative would suffer if the authors discussed internal states more as a theoretical and methodological construct, rather than as a real biological phenomenon.

Authors: Thank you for this comment. By internal states, we mean latent variables (hidden and/or unmeasurable) that influence behavior. Internal states vary over time, on a trial-by-trial basis, and are not discrete. Based on the reviewer’s comment, we now appreciate that the term “states” is also used to describe discrete events (e.g. planning, moving) to partition behavior. This is not what we mean when we use the term “states”. In systems theory, states are variables (often latent) that carry “memory” of a system that influences behavior when provided an input stimulus. That is, The behavior output is a function of both the stimulus and state. Since we construct state-space models, we used states in the context of systems theory. Indeed our two states represent the accumulation of past events (e.g. past perturbations and past errors), i.e., memory, which we show improved prediction of behavior (e.g. RT). To address the reviewer’s concern, we now use the suggested language that internal states are our methodological constructs of “memory” that we believe influence behavior (page 3 **“These factors are commonly represented as a**

methodological construct of memory called internal states.”). We thank the reviewer for this constructive critique.

2. There is a considerable literature on the role of signal variability in noninvasive human imaging that could potentially be relevant , e.g.

Garrett, D. D., Samanez-Larkin, G. R., MacDonald, S. W., Lindenberger, U., McIntosh, A. R., & Grady, C. L. (2013). Moment-to-moment brain signal variability: a next frontier in human brain mapping?. *Neuroscience & Biobehavioral Reviews*, 37(4), 610-624.

McIntosh, A. R., Kovacevic, N., & Itier, R. J. (2008). Increased brain signal variability accompanies lower behavioral variability in development. *PLoS computational biology*, 4(7), e1000106.

Authors: We agree that including the observations of signal variability from noninvasive human imaging studies would be valuable. We have added these points to the introduction:

- page 3 **“However, there is emerging evidence that this variability is purposefully orchestrated by the brain to facilitate learning and adaptation [McIntosh 2008, Garrett 2013].”**

3. There is also a large literature on dynamic states in human imaging, e.g.

Lurie, D. J., Kessler, D., Bassett, D. S., Betzel, R. F., Breakspear, M., Kheilholz, S., ... & Calhoun, V. D. (2020). Questions and controversies in the study of time-varying functional connectivity in resting fMRI. *Network neuroscience*, 4(1), 30-69.

Authors: We agree and have added relevant studies that use noninvasive human imaging to our introduction:

- page 5, **“These studies report the occurrence of co-varying behavioral and neurological variability [McIntosh 2008, Garrett 2013, Lurie 2020]”**

4. Why standard deviation as opposed to the coefficient of variation, which takes mean scaling differences into account?

Authors: We chose to quantify variability using the standard deviation instead of the coefficient of variation for Figure 1d because it is a statistic of variability. We acknowledge that keeping the standard deviation as it incorporates the mean, but the mean is also an important differentiator of variability across subjects.

5. My main methodological concern is the arbitrary (mean) splitting of participants into top and bottom performers. This choice is unnecessary and potentially suboptimal as it transforms an inherently continuous variable into a categorical one. All the analyses could then be performed using correlations (with bootstrapping) rather than ANOVAs.

Authors: We thank the reviewer for this comment. We agree and have redone our analyses using correlations instead of splitting subjects into groups. This has changed the following results:

- Figure 1d. We found that STD of SE significantly correlated with session performance.
- Figure 3 with accompanying results on pages 13-15. We found the same general relationships we previously discussed in that the weight of the error state significantly correlated with session performance. We also found a significant correlation between the weight of speed in RT and session performance (Supplementary Figure 6c).
- Figure 5 c,e with accompanying results on pages 18–20. We found a significant correlation between the encoding strength and session performance for both the error state and perturbed state. This suggests that higher performance may be led by neural power that modulates with the internal states.
- Connectivity results between pages 20–23. We found regions whose encoding strength modulates with internal states are also functionally connected. But the strength by which a pair of regions covaries is also dependent on session performance (Supplementary Figure 9). For example, connections that correspond to higher performance are shown in Figure 6d-e.

6. Connectivity estimated as correlated power. Why not use an established method that focuses on coherence/synchrony or on correlating amplitude envelopes?

Authors: There are several methods for computing functional connectivity. We chose to start with the simplest and most widely used metric for connectivity: correlation. Other methods such as coherence/synchrony or amplitude envelopes would not work here because clusters varied in size (time and frequency).

7. Is there any cross-validation for the fitted internal states model? The authors compare their model with a pure linear model and show that the former is better, but is there any way to demonstrate that the fitted model helps to predict future values of RT?

Authors: Cross-validating our internal state models proved difficult, as the internal states are history-dependent. However, since the focus of our paper was on how internal states influence behavior as opposed to trying to predict behavior, our final models were fitted using all data. To address your point, we implemented a 10-fold cross-validation using the internal states previously fitted and found that adding internal states significantly improved the model's ability to predict RTs (paired-sample t-test: $t(9)=2.45$, $p=0.037$) but not SE (Supplementary Figure 7c). Adding internal states did improve the model's ability to predict RT and SE for all but two subjects.

8. "First, a Notch filter was applied using notches located at the fundamental frequency

of 60 Hz with bandwidth at the -1 dB point set to 3 Hz". The word "notch" does not need capitalizing. Also, did the authors include filters at subsequent harmonics of 60Hz?

Authors: Thank you for this comment. The filters did include higher harmonics of 60 Hz. The methods have been updated to make this clear (page 47, "First, a notch filter was applied to the raw voltage data using notches located at the fundamental frequency of 60 Hz and its higher harmonics with the bandwidth at the -1 dB point set to 3 Hz.").

9. The authors assign regions to frequency bands, but does this method account for the fact that the power spectrum has a $1/f$ shape and low frequencies will naturally tend to have greater power than higher frequencies?

Authors: Again, thank you for this comment. Indeed, this method does account for the reviewer's concern since each frequency bin was independently normalized using the z-score across the entire session. Regardless, the non-parametric cluster statistic is invariant to these problems.

10. The other major methodological concern is how network specificity is established for the reported effects (e.g. localizing states to DAN or DMN), Namely, the comparison was clearly made visually/qualitatively, but the different networks all have different size, spatial coverage, etc. This means that some networks are more likely to get "hits" than others. The authors should confirm that each network is enriched for a particular state by computing the mean in the network and then comparing this to a null distribution where network labels are permuted.

Authors: Our analysis was constrained by the coverage of the implanted depth electrodes across our 10 subjects (seen in Fig 4a). Electrodes were not assigned to networks visually but rather using a rigorous method based on an electrode localization algorithm. The coordinate of each electrode relative to their MRI using an electrode localization method established by Stolk et al. 2018, by fusing the subject's MRI (pre-processed using Freesurfer) with their CT (which contains the coordinates of the electrodes). From there, each electrode was separately mapped to an atlas label (such as intraparietal sulcus L using Destrieux atlas [Destrieux et al. 2010]) and a network label (such as DAN using Yeo atlas [Yeo et al. 2011]) using established the cortical atlases mentioned. This process is described in detail on page 48 of our manuscript. Across all subjects, we had 512 contacts in 40 unique region labels that mapped to 7 networks viable for analysis. These 512 electrodes are mapped to the networks as seen in panel a (below). We acknowledge that our results could be skewed by the disproportionate coverage of electrodes across the main large-scale brain networks. Despite this, we still found a large proportion of electrodes in the underrepresented network of DAN to be encoding the error state seen in panel b (below).

Reviewer #3 (Remarks to the Author):

The main topic addressed by this paper is to examine how the brain represents internal states during motor processing, and to reveal the neural substrate of these patterns. Specifically, this paper utilizes state-space models to link internal states (past errors and perturbations) with motor performance and neural activity, and to identify variations in performance across subjects and trials. While performing a goal-directed center-out reaching task, ten human subjects' electrophysiological data and behavioral data were collected. The paper concluded that large-scale brain networks in the dorsal attention and default networks reflect the neural basis of strategy to regulate movement variability through internal states (past error and perturbation states, respectively) to improve motor performance.

Overall, the idea of examining the neural basis of internal states in regulating an individual's motor performance is interesting. The paper is well written and has many innovative ideas. However, my enthusiasm for the paper is lowered the impact of the neural data analyses is rather limited. The neural power analyses, which found signals at particular frequencies that correlated with aspects of behavior, were not described in a way that lets the reader make specific inferences about the functional contributions of those regions to behavior. The synchrony analyses were improved in this way, but still, there was not enough detail to make a compelling case about the functional or mechanistic roles of particular regions or networks to support motor processing. Further, I have significant issues with the methods and data analysis that the authors presented.

Authors: We thank the reviewer for their positive feedback and helpful comments. In this revision, we provide a point-by-point response to each of the reviewer's specific comments and indicate where we revised the manuscript.

Specific critiques:

1. The behavioral data were not presented with enough detail for me to understand the differences across good versus bad subjects, as well as to explain differences in the subjects that the model fit well versus poorly. I would suggest expanding figure 1 and the related sections of the text to separately analyze and interpret the behavioral characteristics of the group data for the subjects with different levels of absolute performance and for those who the model did or did not fit well.

Authors: We thank the reviewer for their concern. We have since removed grouping subjects by performance and replaced it with the spectrum of subject performance. We believe Figure 1d shows that good performers have less variable RT and SE compared to bad performers, the latter of which is statistically supported. In our revised manuscript, we have also added detailed statistics that confirmed subject differences in RT and SE:

- **Page 9, "Figure 1d shows how subject's variability of RT (top) and SE (bottom) is related to their performance. Specifically, we found that**

subjects with higher session performance had less variable SE (Pearson's correlation: $r=-0.88$, $p=7.42 \times 10^{-4}$)."

We used our model to examine the relationship between session performance and conditions/states, via how subjects weighted model variables. For example, we go into detail about the relationship between the internal states and session performance in Figure 3. Supplementary Figure 6 also shows the comparison between trial conditions and session performance. The only other significant relationship was between the weight of speed, where higher performance correlated with larger weights on speed (Pearson's correlation: $r=0.91$, $p=2.36e-04$), which is brought up in the Results:

- Page 15, **"Another key strategy was speed, where larger magnitudes of weights on speed correlated with high session performance (Supplementary Figure 6c)."**

2. The neural data analyses for figures 4, 5, and 6 are not illustrated with adequate statistical justification. I could not tell which neural signals correlated with behavior in a statistically robust fashion, and there was a lack of adjustment for multiple comparisons. Overall this section of the paper was largely qualitative, which is inadequate. This section of the paper should rigorously show which neural signals, across frequencies, correlated with behavior, and include both within- and across subject comparisons.

Authors: Thank you for your comment. The other reviewers raised similar concerns so we have revised the manuscript as detailed below to more clearly explain our statistical analyses. All of our neural results were derived using an unsupervised and data-driven approach, which were supported by statistical tests at all steps. First, the raw SEEG data was preprocessed into spectral data of normalized power in the time and frequency domains. The spectral data were then sliced into trials, which were then sliced into epochs. Next, we used a non-parametric cluster statistical test [Maris & Oostenveld, 2007] to find windows of time and frequency in which the spectral power correlated with each internal state across trials for each region across the population. This method is agnostic to time and frequency. These windows are called clusters, and each has an associated cluster statistic (Figure 4c). Finally, we corrected for multiple comparisons (one comparison for each brain region for each epoch for each internal state) using a false-discovery rate. What we report in our paper are only the positive results (see Tables 2 and 3 for statistics from nonparametric cluster analysis). As a result, this approach narrowed down our scope from 40 possible regions to just 12 for the error state and 14 for the perturbation state.

Based on this data-driven approach, we noticed that the error state was populated by regions in DAN while the perturbation state was populated by regions in DN (Figure 5a-b). If anything, our approach was extreme in that we used all trials, all regions, all frequencies, etc. We reinforced these results to also show that the strength of which subjects encoded the internal states (via the average power in a cluster) depended on their session performance (Figure 5c-f).

Once the clusters (windows of time and frequency) were statistically identified, we calculated the average power in each cluster for each trial (visualized in Figure 4b). This gave us a signal of how the power in the cluster varied over trials for each region. This naturally led us to run a connectivity analysis between these signals across regions, via Pearson's correlation, for each subject. The number of correlations for each subject varied based on the number of regions that were identified using the non-parametric cluster statistic. But, to minimize the number of statistical tests, we only ran correlations within each internal state result. That is, we ran connectivity analysis for regions related to the error state and perturbed state separately.

Accordingly, based on the reviewer's feedback, we have updated figures and revised the text to make it more apparent that our results were data-driven and supported by statistics.

- Page 16–17:

- “To identify the neural correlates amongst these regions with an **unsupervised and data-driven method**, we used a non-parametric cluster statistic [36] (see Methods) between the spectral data of each region and each internal state across the population (Fig. 4b). This method finds windows of time (relative to epoch onset) and frequency (between 1 and 200 Hz) in which the spectral power of a region significantly correlates with the internal state across trials as demonstrated in Fig. 4c. **The statistic provided us with two sets of neural correlates, one for each internal state.**”
- “Our modeling results showed that subjects weighed internal states differently, primarily based on their session performance. We suspected that this would be reflected in the brain for each subject by how well these regions encoded the states through the strength of their neural correlates. **The degree to which a subject encodes an internal state in a region, which we called the encoding strength, was quantified by correlating the average power within the time-frequency window (from the population statistic) to the state on a trial-by-trial basis (see Methods). Fig. 4d shows an example of how the encoding strength is obtained for a channel in subject 6 using the neural correlate from Fig. 4c.**”

3. The analysis described in Figure 2 was hard to understand because it was focused so much on just subject 6.

Authors: We chose to focus the majority of Figure 2 on subject 6 to help the reader gain intuition about the data and have revised the text to make this point clear. The qualities we described for subject 6 were found across all subjects, whose equivalent figures are given in Supplementary Figures 1–5. We have

subsequently revised the results related to Figure 2 to make their implications more generally applicable beyond subject 6.

- Pages 10—11 **“Though we used subject 6 to demonstrate the intuition behind our model, our general observations are applicable to the population, who are shown in Supplementary Fig. 1 – Supplementary Fig. 5.”**
- Page 11, **“Conditions accounted for the trial-by-trial changes and states accounted for the gradual changes across all subjects (Supplementary Figure 7).”**
- Page 11, **“With the exception of RT for subject 1, we consistently found a significant relationship between the estimated and observed RT (Supplementary Fig. 1) and SE (Supplementary Fig. 2) for all subjects.”**
- Page 12, **“In RT, we also observed states that monotonically increased over trials for subjects 4, 5, 7, and 10 (Supplementary Fig. 3). This state reproduced subjects progressively reacting slower due to fatigue (Supplementary Fig. 1)”**
- Page 12, **“Based on the structure, we can also see what influenced the RT states to be monotonic for subjects 4, 5, 7, and 10. Supplementary Figure 5 shows that the primary contributor was the perturbed state.”**

4. The behavioral data in Figure 2 are hard to interpret because panels B-D are visually very dense. Panel E seems to indicate that the model fit quality is bimodal, with a cluster of four subjects who have four model fits and others who show rather good model fits. Does this undermine the results?

Authors: The bimodality of Figure 2e does not undermine the results. This stems from the fact that models were fit for subjects which led to varying optimal model performance. We discuss the implications of these observations to our results:

- Page 13, **“For example, the goodness-of-fit for subject 6 (black) improved by 16%. However, the model structure fits some subjects better than others. Specifically, subjects 1–4 had outlying model performance for RT (Figure 2e(left)) compared to other subjects. This is notable because these subjects also had the lowest session performance. Because their model performance was low prior to adding internal states, this indicates that their RT varied by factors other than speed and direction which could also explain their low session performance. Despite this, adding internal states also improved the deviance (Supplementary Figure 7b) and 10-fold cross-validation using the fitted internal states for RT (Supplementary Figure 7c). This, in addition to the fact that adding internal states improved their model fit and the inlying goodness-of-fit of their SE model still merits the validity of interpreting their RT model”**

5. Figure 4C is hard to interpret because the statistic is calculated based on the correlation but the colors reflect a different value, mean power. It would be much easier to interpret this plot if the same value was plotted as indicated in the statistic.

Authors: We agree that the original version of Figure 4 was difficult to interpret. We now use Figure 4 to help the readers understand our data-driven approach using non-parametric cluster statistics to identify windows of time and frequency in the spectrum that correlated with internal states. Figure 4 now includes spectrums (Fig. 4b) as a qualitative example as well as the t-statistic (Fig. 4c) as a quantitative example.

6. Figure 6 is hard to interpret, is there a more intuitive or data-driven way to order the labels around the plots?

Authors: We feel that Figure 6 is similar to how connectivity is displayed in other literature. However, we added panels to Figure 6 (a-c) to help explain the method we used to arrive at the connectivity results. The ordering of the labels is data driven in that the regions are ordered by when their cluster appears during the trial (which follows Tables 2 and 3).

- Page 21, **“We quantified functional connectivity by correlating the average power within the time-frequency window, used for encoding strength, on a trial-by-trial basis for each channel per subject (Supplementary Figures 6a and 6b).”**
- Page 21, **“For example, subjects with high performance will exhibit higher subject connectivity strength between key regions (Figures 6c).”**

7. The analysis on Page 20, Line 30 and Fig 6 could benefit from a more quantitative comparison between network connectivity during “error state” and “perturbed state” rather than qualitative comparisons. There are various quantitative measures that the authors might use for this network comparison

Authors: The network connectivity results were derived from quantitative measurements. We’ve elaborated in Figure 6 to show how we quantitatively arrived at our connectivity result. We also added population connectivity in Supplementary Figure 8, though between subject variability was the focus of our paper and not population. We also added Supplementary Figure 9 to showcase all of the performance-modulated networks and not just the ones that lead to higher performance (seen in Figure 6d-e).

8. Perhaps it is a minor point, but I found the conclusion of the paper confusing. How are the results relevant for the type of behavioral data that could be collected from smartphones? There might be an interesting idea here but it needs more length to spell out.

Authors: We agree that our conclusion could be made more straightforward. We have since edited to make our point clearer.

- Page 32, **“Our results raise the possibility that the underlying history of measured behaviors could be used to make inferences about a person’s brain state without needing to collect electrophysiological data, saving time and money in the health field for personalized medicine [81] or business ventures such as sports [82]”**

9. Please explain in the text that discuss in the text why was it was valid for the subjects to be divided based on a median split of RT? Some “top performers” may be doing relatively poorly in the task.

Authors: Subjects were initially divided into “top” and “bottom” performers based on the average session performance of the population, which was 51%. After much consideration, we’ve decided to broaden our analysis by analyzing subjects as a spectrum based on their session performance instead of binary groups. This has changed the following results:

- Figure 1d. We found that STD of SE significantly correlated with session performance.
- Figure 3 with accompanying results on pages 13-15. We found the same general relationships we previously discussed in that the weight of the error state significantly correlated with session performance. We also found a significant correlation between the weight of speed in RT and session performance (Supplementary Figure 6c).
- Figure 5 c,e with accompanying results on pages 18–20. We found a significant correlation between the encoding strength and session performance for both the error state and the perturbed state. This suggests that higher performance may be led by neural power that modulates with the internal states.
- Connectivity results between pages 20–23. We found regions whose encoding strength modulates with internal states are also functionally connected. But the strength by which a pair of regions covaries is also dependent on session performance (Supplementary Figure 9). For example, connections that correspond to higher performance are shown in Figure 6d-e.

REVIEWER COMMENTS

Reviewer #1 (Remarks to the Author):

In this resubmission, the authors attempt to show that internal states account for behavior in a motor task. Unfortunately, they failed to address my previous concerns. The persistent statistical issues throughout the manuscript prevent interpretation of the findings.

The neural analysis methods utilized -- including non-parametric cluster analysis -- are clear, however, they still do not support the claims made by the authors. The authors state

"Specifically, the Dorsal Attention Network (DAN) and Default Network (DN) were linked to the error and perturbed state, respectively" (page 6)

"Remarkably, we then found that these internal states were linked to encoding in large-scale brain networks, DAN and DN, respectively" (page 25)

"For the first time, our results implicate DAN as a network as encoding tracking history" (page 26-27)

"We also found that the DAN encoded error history throughout the trial using persistent activity in frequencies above 100 Hz" (page 27)

To make these claims, the authors must perform direct tests between DAN and another network(s), and across different frequency bands. If one network shows a significant effect and another network does not show a significant effect (as determined by the cluster analysis), this does not mean that there is necessarily a difference between the two networks. DAN could show an effect at $p = 0.04$ and DMN could show an effect at $p = 0.10$, but the two networks need not differ from each other. This inference problem is explained in Makin & Orban de Xivry (2019) "Ten common statistical mistakes to watch out for when writing or reviewing a manuscript." *ELife* ("Interpreting comparisons between two effects without directly comparing them").

The authors make multiple references to figures to support their claims. Figures are for visualization, but statistics are required to make any/all claims. These are absent from the manuscript.

The authors added degrees of freedom (DFs) and test statistics to the behavioral results, but the DFs do not appear to be correct. The authors report ANOVAs with DFs of (1,1296), (3,1296) and (5,1296). Based on the levels and factors, the DFs should be (1,9) for main effect of speed; (3, 27) for main effect of

direction and (3,27) for the interaction between speed and direction (there is also a lack of standard terminology, e.g. "main effect" and "interaction," to communicate the specific tests being performed).

As defined in the manuscript, connectivity is performed by correlating all pairs of electrodes, yet the authors make reference to different regions/networks, which presumably do not include all electrodes. The number of statistical tests performed (based on the response letter) is quite large and it is unclear whether any corrections were applied.

The authors continue to focus on subject 6 with little justification and refer the reader to supplemental figures for the full set of data, which again must be supplemented by statistics. The connectivity correlation shown in Figure 6 is clearly driven by the fact that only three subjects are included and one subject is an outlier in comparison to the other two. In general the correlation results (the majority of the manuscript) seem likely to be driven by outliers given the small sample size.

Frequency band selection is still not defined. The authors state "as commonly defined in literature," however, one or more citations must be provided, as different definitions exist in the literature.

Reviewer #2 (Remarks to the Author):

The authors have comprehensively addressed my concerns and I recommend publication.

Reviewer #3 (Remarks to the Author):

The authors have done a very good job responding to my comments and made a number of improvements to the manuscript. I think the paper is significantly improved and suitable for publication.

My only remaining comment is that I think the authors should clarify the text and caption related to Figure 6C. It was a bit hard to understand how this figure related to the rest of the analyses and how this figure was generated. I eventually figured out how they generated this figure, but they should clarify the sample size that contributed to this analysis and how this sample was generated. This analyses around Figure 6 are mentioned in the abstract, so I think it is especially important for the relevant analyses to be explained clearly.

Reviewer #1 (Remarks to the Author):

In this resubmission, the authors attempt to show that internal states account for behavior in a motor task. Unfortunately, they failed to address my previous concerns. The persistent statistical issues throughout the manuscript prevent interpretation of the findings.

Authors: Thank you for your help and patience as we continue to correct our manuscript to properly adhere to the rigorous statistical standards expected for any scientific work. Based on your comments, we decided to consult a statistician about our analysis methods. We believe the quality of our findings has greatly improved as a result and assure the review that we have now properly addressed any remaining concerns that you may have about the statistical validity of the manuscript. The manuscript is marked based on items that were removed (red text with strikethrough) or added (blue text).

1. The neural analysis methods utilized -- including non-parametric cluster analysis -- are clear, however, they still do not support the claims made by the authors. The authors state

"Specifically, the Dorsal Attention Network (DAN) and Default Network (DN) were linked to the error and perturbed state, respectively" (page 6)

"Remarkably, we then found that these internal states were linked to encoding in large-scale brain networks, DAN and DN, respectively" (page 25)

"For the first time, our results implicate DAN as a network as encoding tracking history" (page 26-27)

"We also found that the DAN encoded error history throughout the trial using persistent activity in frequencies above 100 Hz" (page 27)

To make these claims, the authors must perform direct tests between DAN and another network(s), and across different frequency bands. If one network shows a significant effect and another network does not show a significant effect (as determined by the cluster analysis), this does not mean that there is necessarily a difference between the two networks. DAN could show an effect at $p = 0.04$ and DMN could show an effect at $p = 0.10$, but the two networks need not differ from each other. This inference problem is explained in Makin & Orban de Xivry (2019) "Ten common statistical mistakes to watch out for when writing or reviewing a manuscript." *ELife* ("Interpreting comparisons between two effects without directly comparing them").

Authors: We thank the reviewer for pointing this mistake out! The reference offers the following solutions:

"Researchers should compare groups directly when they want to contrast them (and reviewers should point authors to Nieuwenhuis et al., 2011 for a clear explanation of the problem and its impact). The correlations in the two groups can be compared with Monte Carlo simulations (Wilcox and Tian, 2008). For

group comparisons, ANOVA might be suitable. Although non-parametric statistics offers some tools (e.g., Leys and Schumann, 2010), these require more thought and customisation.”

To determine whether there is a statistically significant relationship (p -value < 0.5) between the specific large-scale network in a given frequency band and the internal states per trial, we correlated (Spearman’s correlation, consistent with the type of correlation used for the non-parametric cluster statistic) the average power of the cluster for each trial aggregated over all subjects from each region in either DAN or DN that we found to encode each internal state, which we also visualized through scatter plots:

The top row represents the error state and the bottom row represents the perturbed state, where j represents the number of regions plotted. The left plot displays DAN and DN aggregated, the middle plot displays only the regions in DAN and the right plot displays only the regions in DN. We found all the correlations to be significant (p -value < 0.5), *i.e.*, DAN and DN encode both internal states. What we can see is that, of the total regions in DAN+DN, the majority of regions encoding the error state were in DAN (70%) and the majority of regions encoding the perturbed state were in DN (~77%). This supports the claim in our manuscript in which the error state is primarily encoded by DAN and the perturbed state is encoded primarily by DN.

Additionally, we were also interested in the relationship between the specific regions in each network and the internal state. Thus, we also examined the correlation (Spearman’s correlation) between internal states and the regions within each network aggregating subjects:

The figure above represents the error state, where each plot represents the average power of a cluster for each trial across all subjects from the region in the title (n represents the number of subjects aggregated in the scatter plot). The top row represents the regions from DAN that encoded the error state (j refers to the specific number of regions) whereas the bottom row represents the regions from DN. We found each region to be significantly correlated (p-value < 0.5) with the internal state. Using the correlation value as a measurement for how well a region encoded the internal state, we found that the regions in DN (bottom row, median of 0.16) encoded the error state as well as the regions in DAN (top row, median of 0.19) supported through a statistical test (Mann-Whitney U = 39, p = 1).

All regions encoding the perturbed state were significantly correlated (p-value < 0.5) as well (using Spearman's correlation). Similarly, for the perturbed state, the regions in DAN (top row, median=0.17) that also encoded it did so as well as the regions in DN (bottom row, median=0.12) (Mann-Whitney U = 26, p = 0.47).

Again, what we can draw from both of these figures is that more regions in DAN encoded the error state than DN (7 regions in DAN vs. 3 regions in DN) and more regions in DN encoded the perturbed state than DAN (3 regions in DAN vs. 10 regions in DN). Therefore, the ultimate conclusion we can make from our results is not that DAN exclusively encodes the error state but rather that the

majority of regions that encoded the error state were in DAN and the majority of regions that encoded the perturbed state were in DN.

What we cannot draw from these relationships is the varying degree to which subjects actually encode the internal states, i.e. the basis for our claim that encoding scales by performance. Below, we found the correlation between encoding regions with internal states (and their p-value) varied based on subject performance, most easily summarized by aggregating all the regions subjects had recordings from for either DAN or DN, visualized in a scatter plot:

For the error state (left) and perturbed state (right), we found that higher-performing subjects had higher correlation values for both the regions in DAN and DN (where j represents the number of regions aggregated per subject). The correlations also revealed that the higher-performing subjects were more likely to have a significant correlation (p-value < 0.5) than the lower-performing subjects.

To check whether these results were not simply skewed by possible disproportionate coverage of the electrodes to any particular networks, we compared the actual coverage of the networks from population to the coverage found to encode either internal state:

As you can see, the networks from our results do not match the proportions of networks from the population.

With that said, we modified the language in our manuscript to better reflect the more rigorous analyses above regarding results:

- Page 1 (Abstract): “The dorsal attention network **primarily** encoded past errors in frequencies above 100 Hz, suggesting a role in modulating attention based on tracking recent performance in working memory. The default network **primarily** encoded past perturbations in frequencies below 15 Hz suggesting a role in achieving robust performance in an uncertain environment.”
- Page 6 (Introduction): “Specifically, the Dorsal Attention Network (DAN) and Default Network (DN) were linked to the error and perturbed state, **respectively.**”
- Page 18 (Results): “Some regions from other networks also appeared, such as the **angular gyrus right (AG R) and posterior-dorsal cingulate gyrus right (dPCC R) in DN**, the supramarginal gyrus right (SMG R) from the Ventral Attention Network (VAN), as well as cuneus right (Cu R) in the Visual Network (Visual). **However, the majority of regions that encoded the error state were located in DAN.**”
- Page 20 (Results): “**Overall, the majority of regions we found to encode the perturbed state were in DN.**”
- Page 20 (Results): “**This figure also shows regions from other networks, which include the DAN in dark blue, VAN in light blue, Somatomotor in green, and Visual in yellow.**”
- Page 25 (Discussion): “Remarkably, we then found that these internal states were linked to encoding in large-scale brain networks, DAN and DN, **respectively.**”

- Page 26 (Discussion): “We found that our subjects learned the task by monitoring their history of errors across trials to decide where to allocate attention **primarily** through the DAN.”
- Page 28 (Discussion): “Finally, we found that the encoding strength of regions in the DAN and functional connectivity between these regions, **along with the visual network**, scaled based on the subject’s overall performance.”
- Page 29 (Discussion): “We found that regions whose activity correlated with the perturbed state were **primarily** in the DN.”
- Page 30 (Discussion): “Taken together, these findings suggest that high performers react to uncertainty by heightening vigilance through activation and connectivity in DN **in conjunction with the VAN and the visual network.**”

2. The authors make multiple references to figures to support their claims. Figures are for visualization, but statistics are required to make any/all claims. These are absent from the manuscript.

Authors: After carefully reviewing our revised manuscript, we are confident that it contains the appropriate statistics for any/all claims that were made. In particular, in this version of our manuscript, we have added the p-values and sample size to the performance connectivity analysis in Supplementary Figure 8 which was summarized in Figure 6.

3. The authors added degrees of freedom (DFs) and test statistics to the behavioral results, but the DFs do not appear to be correct. The authors report ANOVAs with DFs of (1,1296), (3,1296) and (5,1296). Based on the levels and factors, the DFs should be (1,9) for main effect of speed; (3, 27) for main effect of direction and (3,27) for the interaction between speed and direction (there is also a lack of standard terminology, e.g. "main effect" and "interaction," to communicate the specific tests being performed).

Authors: Thank you for your suggestions. After consulting with Emery Brown, a statistician, we have corrected the reporting of the statistics used on our behavioral analysis to use the correct degrees of freedom:

- Page 8 (Results): “That is, there was a main effect of subject on RT (three-way ANOVA: **F(9,27)=7.24, p=2.85x10⁻³, partial eta²=0.87**) and SE (three-way ANOVA: **F(9,45)=13.17, p=4.09x10⁻¹², partial eta²=0.62**).”
- Page 8 (Results): “Indeed, there was a main effect of speed on RT (three-way ANOVA: **F(1,9)=9.74, p=0.012, partial eta²=0.52**), meaning subjects reacted more quickly for fast trials than slow trials. We also found a main effect between the location of the target and how quickly subjects

reacted (three-way ANOVA: $F(3,27)=4.59$, $p=0.0012$, **partial $\eta^2=0.32$** .)”

- Page 9 (Results): “We found a main effect between the type of perturbation and SE (three-way ANOVA: $F(5,45)=23.23$, $p=9.56 \times 10^{-12}$, **partial $\eta^2=0.71$**), with the exception of unperturbed compared to towards trials regardless of speed (Supplementary Table 10).”

We also modified our results and methods to better communicate the statistical tests being used:

- Page 8 (Results): “That is, there was a **main effect** of subject on RT (three-way ANOVA: $F(9,27)=7.24$, $p=2.85 \times 10^{-3}$, **partial $\eta^2=0.87$**) and SE (three-way ANOVA: $F(9,45)=13.17$, $p=4.09 \times 10^{-12}$, **partial $\eta^2=0.62$**).”
- Page 8 (Results): “Indeed, there was a **main effect** of speed on RT (three-way ANOVA: $F(1,9)=9.74$, $p=0.012$, **partial $\eta^2=0.52$**), meaning subjects reacted more quickly for fast trials than slow trials. We also found a **main effect** between the location of the target and how quickly subjects reacted (three-way ANOVA: $F(3,27)=4.59$, $p=0.0012$, **partial $\eta^2=0.32$**).”
- Page 9 (Results): “However, there was not a **significant interaction** between speed and direction.”
- Page 9 (Results): “We found a **main effect** between type of perturbation and SE (three-way ANOVA: $F(5,45)=23.23$, $p=9.56 \times 10^{-12}$, **partial $\eta^2=0.71$**), with the exception of unperturbed compared to towards trials regardless of speed (Supplementary Table 10).”
- Page 40 (Methods): “To test the **main effects** and **interactions** that subjects and trial conditions had on RT and SE, we constructed a multi-way ANOVA for each.”
- Page 11 (Supplementary): “Supplementary Table 4 | Three-way ANOVA of **main effects and interactions** that subject, speed, and direction have on reaction time. (df = degrees of freedom, SS = Sum of Squares, MS = Mean Squares, EMS = Expected Mean Squares, F = F -statistic)”
- Page 12 (Supplementary): “Supplementary Table 5 | Post-hoc Tukey’s comparison for the **main effect** of subject on reaction time.”
- Page 13 (Supplementary): “Supplementary Table 6 | Post-hoc Tukey’s comparison for the **main effect** of direction on reaction time.”
- Page 14 (Supplementary): “Supplementary Table 7 | Post-hoc Tukey’s comparison for the **interaction** between speed and direction on reaction time.”
- Page 15 (Supplementary): “Supplementary Table 8 | Three-way ANOVA of **main effects and interactions** that subject, type of perturbation (speed, pert.), and reaction time (RT) have on speed error. (df = degrees of freedom, SS = Sum of Squares, MS = Mean Squares, EMS = Expected Mean Squares, F = F -statistic)”
- Page 16 (Supplementary): “Supplementary Table 9 | Post-hoc Tukey’s comparison for the **main effect** of subject on speed error.”

- Page 17 (Supplementary): “Supplementary Table 10 | Post-hoc Tukey’s comparison for the **main effect** of type of perturbation on speed error.”

4. As defined in the manuscript, connectivity is performed by correlating all pairs of electrodes, yet the authors make reference to different regions/networks, which presumably do not include all electrodes. The number of statistical tests performed (based on the response letter) is quite large and it is unclear whether any corrections were applied.

Authors: Prior to performing any additional analysis, we corrected the nonparametric cluster statistic results for multiple comparisons (one comparison for each brain region for each epoch for each internal state) using a false-discovery rate (FDR) of $q=0.015$. We only reported and used the clusters that passed the FDR. For connectivity, to limit the number of correlations performed, the normalized power across trials was averaged across electrodes in the same region for each subject. Hence, the sample size used for correlations within each subject (i.e., between regions for connectivity) was limited to the number of trials they performed.

5. The authors continue to focus on subject 6 with little justification and refer the reader to supplemental figures for the full set of data, which again must be supplemented by statistics.

Authors: We again note that we chose subject 6 as a representative example of the entire dataset:

1. Their performance is around average,
2. The dynamics of their internal states provide key points that aid in interpreting the impact internal states have on behavior for the population,
3. Their neural coverage provided a basis for transitioning from regions to large-scale brain networks using the error state and DAN as an example.

Using the same subject throughout the paper and figures provided consistency for narrative purposes and is justified by statistics stated in the manuscript. We have added the following statements to the manuscript to convey the justification of using subject 6 while conserving the narrative:

- Page 11 (Results): “**To illustrate this, consider as an example subject 6, whose session performance was around average across all subjects.** Figure 2b shows the RT across all trials for this subject, with **their** observed behavior in gray and their estimated behavior in black.”
- Page 11 (Results): “**Continuing with subject 6 as an example,** Figure 2c shows **their** estimated RT separated into the trial conditions (orange) and internal states (purple).”

- Page 11 (Results): “**Generally**, conditions accounted for the trial-by-trial changes and states accounted for the gradual changes across all subjects (Supplementary Fig. 7a).”
- Page 12 (Results): “**Following subject 6**, Figure 2d shows **their** error state (blue) and perturbed state (red) across all trials.”

6. The connectivity correlation shown in Figure 6 is clearly driven by the fact that only three subjects are included and one subject is an outlier in comparison to the other two. In general the correlation results (the majority of the manuscript) seem likely to be driven by outliers given the small sample size.

Authors: First, we want to emphasize that the connectivity results are preliminary results, due to the limited sample size as you pointed out. The main findings of our manuscript are (i) explaining variability with states and (ii) identifying regions that belong to large-scale networks are encoding these states.

We also felt that it was important to be as transparent as possible about the statistics we used to get to our results. Therefore, we now provide the p-values and sample sizes for the performance connectivity analysis (from Figure 6) in Supplementary Figure 8. We also edited our manuscript to emphasize the preliminary aspect of the performance-related correlation results:

- Page 1 (Abstract): “Moreover, these networks were **preliminarily** found to more strongly encoded internal states and be more functionally connected in higher performing subjects, whose learning strategy was to respond by countering with behavior that opposed accumulating error.”
- Page 6 (Introduction): “We also have **preliminary** evidence linking the encoding strength and functional connectivity of these networks back to subject performance and strategy.”
- Page 30 (Discussion): “Our **preliminary** findings suggest that high performers effectively implement their new semantic knowledge about the environment to explore different approaches to prepare for the possibility of future perturbations.”
- Page 28 (Discussion): “Finally, we found that the encoding strength of regions in the DAN and functional connectivity between these regions, along with the visual network, scaled based on the subject’s overall performance. **Though these results are preliminary**, we linked this with observations that subjects have different strategies.”
- Page 32 (Discussion): “**Finally, we want to emphasize that the performance-related results should be taken as preliminary as the sample sizes for these statistics were limited to the number of subjects implanted in the same regions. Hence, studies with larger sample sizes are needed to make any conclusive statements.**”
- Page 54 (Methods): “To identify **preliminarily** pairs of regions whose connectivity encodes the internal states and performance, we related the

subject connectivity strength to session performance for each pair across subjects the Pearson's correlation value (Fig. 6c)."

- Page 54 (Methods): **"Supplementary Figure 8 b,e and Supplementary Figure 8 c,f show the p-values and sample sizes used when calculating the performance connectivity strength."**

7. Frequency band selection is still not defined. The authors state "as commonly defined in literature," however, one or more citations must be provided, as different definitions exist in the literature.

Authors: Thank you for bringing this to our attention. The new manuscript now contains references for each frequency band:

- Page 50 (Methods): "delta (1–4 Hz) [98], theta (4–8 Hz) [98], alpha (8–15 Hz) [98], beta (15–30 Hz) [98], low gamma (30–60 Hz) [99], high gamma (60–100 Hz) [99], and hyper gamma (100–200 Hz) [100]"

98. Prerau, M. J., Brown, R. E., Bianchi, M. T., Ellenbogen, J. M. & Purdon, P. L. Sleep neurophysiological dynamics through the lens of multitaper spectral analysis. *Physiology* 32, 60–92 (2017).
99. Sani, O. G. et al. Mood variations decoded from multi-site intracranial human brain activity. *Nature Biotechnology* 36, 954–961 (2018).
100. Crone, N. E., Sinai, A. & Korzeniewska, A. High-frequency gamma oscillations and human brain mapping with electrocorticography. *Progress in Brain Research* 159, 275–295 (2006).

Reviewer #3 (Remarks to the Author):

The authors have done a very good job responding to my comments and made a number of improvements to the manuscript. I think the paper is significantly improved and suitable for publication.

Authors: We thank the reviewer for their positive reception to our revisions. We addressed their final comment in this latest revision and have indicated where we revised the manuscript. The manuscript is marked based on items that were removed (red text with strikethrough) or added (blue text).

1. My only remaining comment is that I think the authors should clarify the text and caption related to Figure 6C. It was a bit hard to understand how this figure related to the rest of the analyses and how this figure was generated. I eventually figured out how they generated this figure, but they should clarify the sample size that contributed to this analysis and how this sample was generated. This analyses around Figure 6 are mentioned in the abstract, so I think it is especially important for the relevant analyses to be explained clearly.

Authors: Thank you for reviewing our manuscript again. We agree that the sample size is an important point for our preliminary connectivity results. As such, we provided more description about the sample size used for the performance connectivity analysis in the methods:

- Page 54 (Methods): **“Any pair that did not include both a subject above and below average session performance was excluded. We also excluded pairs that were not represented by at least three subjects.”**
- Page 54 (Methods): **“Supplementary Figure 8 b,e and Supplementary Figure 8 c,f show the p-values and sample sizes used when calculating the performance connectivity strength.”**

We also explicitly provide the sample size used to get Figure 6 in Supplementary Figure 8 c,f on page 6 of Supplementary.

Reviewers' comments:

Reviewer #1 (Remarks to the Author):

Although the authors have revised their manuscript, my concerns regarding statistics and analysis approaches remain. The majority of reported findings are not supported by direct tests or group statistics, instead the authors present qualitative comparisons and/or rely on individual subject data, severely limiting the interpretability and generalizability of the findings.

Review 1 Comment: Although the authors have revised their manuscript, my concerns regarding statistics and analysis approaches remain. The majority of reported findings are not supported by direct tests or group statistics, instead the authors present qualitative comparisons and/or rely on individual subject data, severely limiting the interpretability and generalizability of the findings.”

Authors: In our responses over three revisions, we consulted an expert statistician (PhD in Statistics who also conducts studies in neuroscience), and carefully revised our statistics. We adapted our figures and ran multiple analyses to address reviewers’ concerns over the year (including direct tests and group statistics). We also clarified that none of our results were based on qualitative comparisons. We had originally highlighted one subject throughout in figures to show raw data and to help better understand what our models capture; and made this clear that it was just an example and that rigorous quantitative comparisons were performed throughout the study and reported. The interpretability and generalizability of our findings has been clarified in the final revision.

After the second cycle, reviewers 2 and 3 accepted our manuscript for publication, while reviewer 1 still had two concerns. The first concern was about the validity of our claims. We ran the additional analysis as requested and have added this to our revised manuscript along with an additional figure to support our claims. The second concern was about clarifying the use of an example to illustrate general observations, which we have subsequently addressed.

We believe the quality of our findings has greatly improved as a result and assure the reviewer that we have now properly addressed any remaining concerns.